# Chemogenetic attenuation of cortical seizures in nonhuman primates

**Naohisa Miyakawa** [1] ✉, **Yuji Nagai** [1], **Yukiko Hori** [1], **Koki Mimura**[1],
**Asumi Orihara** [1,2], **Kei Oyama**[1], **Takeshi Matsuo**[3], **Ken-ichi Inoue**[4],
**Takafumi Suzuki**[5], **Toshiyuki Hirabayashi**[1], **Tetsuya Suhara**[6], **Masahiko Takada**[4],
**Makoto Higuchi** [1], **Keisuke Kawasaki** [7] **& Takafumi Minamimoto** [1] ✉

Epilepsy is a disorder in which abnormal neuronal hyperexcitation causes several types of seizures. Because pharmacological and surgical treatments occasionally interfere with normal brain function, a more focused and on-demand approach is desirable. Here we examined the efficacy of a chemogenetic tool—designer receptors exclusively activated by designer drugs (DREADDs)—for treating focal seizure in a nonhuman primate model. Acute infusion of the GABA$_A$ receptor antagonist bicuculline into the forelimb region of unilateral primary motor cortex caused paroxysmal discharges with twitching and stiffening of the contralateral arm, followed by recurrent cortical discharges with hemi- and whole-body clonic seizures in two male macaque monkeys. Expression of an inhibitory DREADD (hM4Di) throughout the seizure focus, and subsequent on-demand administration of a DREADD-selective agonist, rapidly suppressed the wide-spread seizures. These results demonstrate the efficacy of DREADDs for attenuating cortical seizure in a nonhuman primate model.

Epilepsy is a brain disease in which a cluster of neurons sometimes induces episodes of abnormal excitation, called epileptic seizures. In severe cases, epileptic discharges can spread to a broad area of the brain, causing loss of consciousness and/or repeated involuntary movements of the body (clonic seizure). Patients diagnosed with drug-resistant (or refractory) focal epilepsy might be considered for surgical resection of the pathogenic area[1]. Frontal lobe epilepsy (FLE) is the second most common type of epilepsy treated by surgical resection, with temporal lobe epilepsy (TLE) being the most common[2,3]. Because the epileptic focus in FLE sometimes involves cortical areas related to motor, speech, and executive functions, resection for drug-resistant FLE surgery is carried out conservatively, resulting in higher unsatisfactory outcome rates (45–60%) than those reported for resection in

TLE[3–6]. Despite ongoing efforts to develop alternative treatments, such as deep brain stimulation[7,8] and vagus nerve stimulation[9], results have been unsatisfactory. Thus, a great need for alternative and effective treatments remains for drug-resistant epilepsy, especially when foci are difficult to treat with conventional surgical approaches.

The chemogenetic tool termed DREADDs (designer receptors exclusively activated by designer drugs) affords a minimally invasive means of reversibly and selectively controlling the activity of a target neuronal population[10,11]. This is accomplished via genetically introduced artificially designed receptors that are not activated by any endogenous ligands, and instead are designed to be activated by biologically inert actuator drugs administered on-demand[10–12] (but can also be activated by clozapine, an antipsychotic drug[13]). DREADDs have

[1]Department of Functional Brain Imaging, National Institutes for Quantum Science and Technology, Chiba, Japan. [2]Department of Neurosurgery, Graduate School of Medical and Dental Sciences, Tokyo Medical and Dental University, Tokyo, Japan. [3]Tokyo Metropolitan Neurological Hospital, Tokyo, Japan. [4]Systems Neuroscience Section, Center for the Evolutionary Origins of Human Behavior, Kyoto University, Aichi, Japan. [5]Center for Information and Neural Networks, National Institute of Information and Communications Technology, Suita, Japan. [6]Institute for Quantum Life Science, National Institutes for Quantum Science and Technology, Chiba, Japan. [7]Department of Physiology, Niigata University School of Medicine, Niigata, Japan.
✉e-mail: nmiyakawa-ns@umin.ac.jp; minamimoto.takafumi@qst.go.jp

been widely used to modify neuronal activity and behavior in rodents and nonhuman primates (NHPs). Numerous preclinical studies show the utility of DREADDs for treating epilepsy in different circumstances, including cultured neurons in mice[14], focal suppression of hippocampal seizures in pharmacologically[15] and electrically kindled model mice[15,16], suppression of seizures in the amygdala kindled model[17], and attenuation of pharmacologically activated seizures in the rat neocortex[18]. Before chemogenetic approaches can be used to treat human patients, their efficacy and safety must be optimized using NHP models, which are the models that best resemble humans because of the size, complexity, and genetic proximity of the NHP brain to our own[19,20]. However, the effectiveness of DREADDs in suppressing seizures in NHP models has not yet been determined. Additionally, their therapeutic application to human epilepsy poses several challenges. First, the effectiveness and safety (e.g., DREADD-specific cell damage) of the genetic transduction must be evaluated. Second, the DREADD actuator should be reconsidered because CNO—the first-generation actuator for the muscarinic DREADD used in all of the above-mentioned studies—has been shown to be reverse-metabolized into clozapine[13], which has potential off-target effects on endogenous systems[12]. Third, the anti-seizure effect of DREADDs must be assessed along with the spatial spread of the seizure because epileptic symptoms are known to become much more serious and difficult to treat in wide-spread (generalized) seizures.

To address these issues and acquire a proof-of-concept for the chemogenetic treatment of seizures in NHP, here we used an NHP model of FLE, which represents a surgically refractory case. We used clinically accessible monitoring systems, including positron emission tomography (PET) to confirm DREADD expression and video-electrocorticogram (ECoG) to monitor the seizures. We introduced an inhibitory designer receptor (hM4Di) into the forelimb region of the primary motor cortex (MI) in two monkeys. Then, we pharmacologically induced acute seizures in the hM4Di-expressing cortical area, which resulted in recurrent cortical discharges with hemi- and whole-body clonic seizures. Next, we demonstrated that administering deschloroclozapine (DCZ), a highly selective DREADD actuator[12], rapidly attenuated the large clonic seizures and the spreading of cortical

seizures. These results suggest that DREADDs might be a future therapeutic option for treating epilepsy in humans

## Results

### Virally mediated transduction of an inhibitory DREADD into monkey MI neurons

To validate the effect of DREADDs in suppressing cortical seizures in primate brains, we made an experimental paradigm (Fig. 1a) in which we (1) focally expressed hM4Di in the monkey brain, as confirmed via in vivo PET imaging, (2) pharmacologically induced acute focal seizure within the hM4Di expression site, (3) recorded cortical seizure activity using ECoG electrode(s) covering a wide region of the brain, and (4) monitored clonic seizures with video cameras.

Two macaque monkeys underwent a craniotomy to expose the central sulcus. To focally express hM4Di, we used a mosaic adeno-associated virus vector (AAV2.1-hSyn-FLAG-hM4Di-IRES-AcGFP) for neuron-specific expression of hM4Di and a fluorescent marker. We injected the vector in multiple locations within the anatomically defined hand/arm region of the left MI (nine injections in an ~3 × 3-mm area; Fig. 1b), which later became the target location for epileptogenic drug injection. Six weeks after the virus injection, DREADD expression was visualized in vivo via PET imaging with the DREADD-selective radioligand [11]C-DCZ[12]. We observed a prominent PET signal in both monkeys, indicating successful hM4Di expression at the target region (Fig. 1c, d and Supplementary Fig. 1a), which was confirmed by post-mortem immunohistochemical staining with an antibody for the co-expressed AcGFP (Fig. 1e and Supplementary Fig. 1b). We verified the chemogenetic function of hM4Di expressed in the MI by testing the grasping ability of a monkey (monkey #2; Table 1). Systemic administration of the DREADD actuator DCZ (0.1 mg kg$^{-1}$, intramuscular injection) significantly and reproducibly impaired food retrieval using the contralateral (right) hand—but not the ipsilateral (left) hand—compared with vehicle control treatment (paired t-test; contralateral: $p = 0.0097$; ipsilateral: $p = 0.74$; Supplementary Fig. 2). These results indicate that activation of hM4Di with DCZ attenuated the activity of neurons in the hand region of the MI, and that DCZ alone did not affect the motor ability of the animal in general.

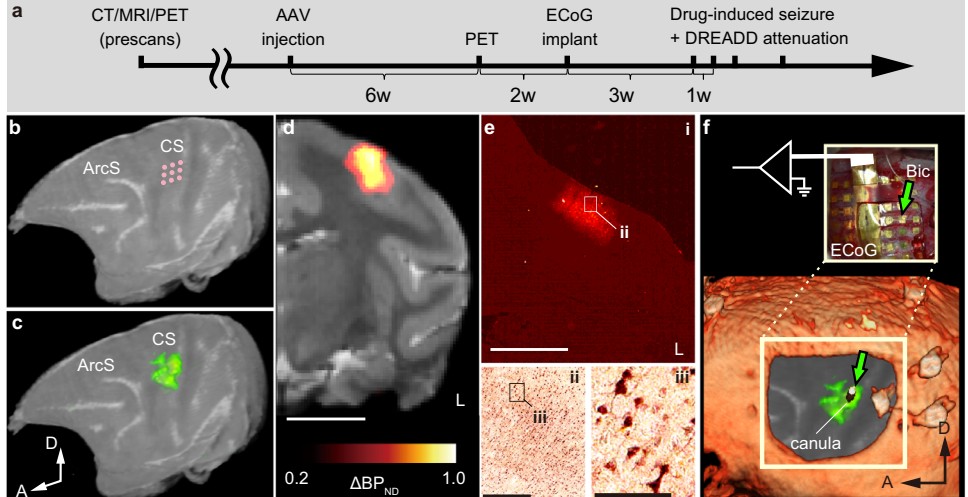

**Fig. 1 | Study design and timeline. a** Study timeline. **b** Schematic showing the location and spatial configuration of the AAV vector injections superimposed on a 3D-rendered MR image of the target monkey brain. **c** Location of the actual hM4Di expression visualized by [11C]DCZ-PET, co-registered on the MR image (arbitrary scale). **d** The co-registered PET/MR image in a coronal plane. **e** Immunohistochemical confirmation of the gene expression using an anti-GFP antibody. **e-i** is converted to pseudo color. Frames in **e-i** and **e-ii** depict the respective area visualized in **e-ii** and **e-iii**. **f** Images of a skull (CT), cortex (MR), and hM4Di expression (PET) were co-

registered to show the relative spatial configuration of the bicuculline infusion site and the target brain region expressing hM4Di. The dummy guide cannula (green arrow) visualized with CT indicates the position of bicuculline infusion. A 64-channel ECoG electrode was placed epidurally and partially beneath the skull. A transparent artificial dura with a manipulating window was slid beneath the skull to hold down the electrode in place. Scale bars: **d** 10 mm; **e-i** 5 mm; **e-ii** 500 μm; **e-iii** 100 μm. Data were from monkey #1 and similar observations were made on monkey #2 (See Supplementary Fig. 1).

**Table 1 | Summary of experimental conditions and setup**

| Subject | MI | AAV injection | Bicuculine | Vehicle | DCZ | ECoG Rec | PET | Grasping | Histology |
|---|---|---|---|---|---|---|---|---|---|
| Monkey #1 | Left | 13.5 μL, $3.5 \times 10^{12}$ cp/mL | 2 | 1 | 2 | 64 ch | ✓ | - | ✓ |
| | Right | - | - | - | - | - | | - | |
| Monkey #2 | Left | 13.5 μL, $3.5 \times 10^{13}$ cp/mL | 4 | 3 | 4 | 64 ch | ✓ | 5 | ✓ |
| | Right | - | 2 | 1 | 2 | 64 ch | | 5 | ✓ |

A check indicates the subject was used in the experiments. The numbers in the Bicuculline column indicate the number of injection sessions into the left or right MI. The numbers in the Vehicle and DCZ columns indicate the number of sessions of each treatment. The numbers in the Grasping column indicate the number of sessions in which the monkey was tested on the modified Brinkman board task.

*MI* the primary motor cortex, *AAV* adeno-associated virus, *cp* capsid particles, *ECoG* electrocorticography, *PET* positron emission tomography.

## A primate model of cortical seizures

After the PET imaging, we surgically placed epidural film-electrode arrays for electrocorticography (ECoG) into the cranial window and held them down with a transparent artificial dura (Fig. 1f). Acute seizures were induced by local injection of bicuculline—a GABA$_A$ receptor antagonist—into the hM4Di-expressing region through a window in the artificial dura (Fig. 1f, green arrows). An example of acute seizure development that was recorded adjacent to the bicuculline infusion site is shown in Fig. 2. Immediately after bicuculline infusion, spectral power lower than 20 Hz initiated an upward trend (Fig. 2b), reflecting epileptic "spikes", "spike-wave complexes", and "multi-spike-wave complexes" (or "poly-spikes") (Fig. 2a-ii, iii; Supplementary Fig. 3a-ii, iii for a longer time period), which are stereotypical of preliminary and mild epileptic seizures[21]. Concurrently, clonic seizures, such as twitching and tremor in the right (contralateral) hands, were also observed (Fig. 2b, bottom). The spectral power then increased intermittently over the frequency range, coinciding with the periods when clonic seizures spread beyond the contralateral arm to parts of the head (e.g., ears, lips, and eyelids) or the entire body (Fig. 2b). This robust increase corresponded to "status epilepticus" (or "sustained multi-spike-complexes") (Fig. 2a-iv and Supplementary Fig. 3a-iv), which is the most severe form of epileptic seizure[21].

## Chemogenetic attenuation of cortical and clonic seizures

After seizures became behaviorally visible, we administered DCZ alone, or in the vehicle followed by DCZ, intramuscularly (Supplementary Table 1). DCZ injection resulted in a rapid decrease in seizure amplitude, disappearance of multi-spike-wave complexes, and concomitant decrease in clonic seizures (Figs. 2a-v, b and Supplementary Fig. 3a-v, b). Although seizure discharges and clonic seizures occasionally recurred after DCZ injection (e.g., ~25 min after the first DCZ injection, Fig. 2b), seizure amplitudes were lower and clonic seizures were fewer compared with what was observed following sham treatments (i.e., vehicle) (Fig. 2c). Moreover, recurring cortical seizures and bodily or bilateral clonic seizures were clearly attenuated after DCZ administration (Fig. 2b, c, yellow arrowhead and Supplementary Fig. 3c). We quantified the onset and strength of the anti-seizure effects for DCZ (six sessions in total from the two monkeys) and compared them with those for the vehicle control (four sessions). At 1 min after DCZ administration, the mean seizure amplitude decreased and remained below the 95th percentile of what it was during the baseline period. In contrast, it did not change after vehicle administration (Fig. 3a). The effect of treatment on seizure amplitudes differed significantly between DCZ and vehicle such that DCZ clearly triggered a drop in amplitude (the 99% credible intervals for seizure amplitudes 3 min after treatment (DCZ or vehicle) did not overlap; Fig. 3b). Furthermore, administration of DCZ, but not the vehicle, rapidly attenuated clonic seizures (convulsions in the body, head, and arms), which remained below the baseline range within 3 min (Fig. 3c). In addition to declining in the right hand, which is controlled by the target left MI cortex (58% decline; Supplementary Fig. 4b left), clonic

seizures also declined in other parts of the body (99% decline, Supplementary Fig. 4c, left). As with seizure activity in the brain, only DCZ significantly affected clonic seizures, again triggering a clear drop in the amplitude (the 99% credible intervals for clonic seizures 3 min after treatment (DCZ or vehicle) did not overlap; Fig. 3d).

We continued to examine the effect of DCZ on both cortical and clonic seizures throughout the 1-h session after the first dose of DCZ. We found that the anti-seizure effects of the first dose lasted at least 20 min for both monkeys. For monkey #1, they were consistently observed for 60 min in both sessions (Supplementary Fig. 5a, red dotted lines). However, for monkey #2, the seizure amplitude gradually returned, and the effects became less clear after 40 min (Supplementary Fig. 5a, red lines). An additional dose of DCZ did not appear to further attenuate seizures in either monkey (second DCZ; Supplementary Fig. 5b). In contrast, DCZ-triggered reduction in clonic seizures consistently lasted for 60 min in both monkeys, although it occasionally weakened in some cases (Supplementary Fig. 5c). As with cortical seizures, the second dose of DCZ had no effect on clonic seizures (Supplementary Fig. 5d). Thus, the shorter duration of the anti-seizure effects observed in monkey #2 was likely due in some way to the differences between the two monkeys (see the Discussion).

We next confirmed that the observed anti-seizure effect of DCZ was DREADD-dependent. When bicuculline was applied to the contralateral MI in monkey #2 (where the DREADD had not been introduced; Supplementary Table 1), administering DCZ did not attenuate cortical seizures; rather, seizures increased significantly after administration (t-test, p = 0.0016; Supplementary Fig. 6), which presumably reflected the natural course of seizure development. These results demonstrate that activation of hM4Di in the seizure focus is effective at suppressing acute cortical seizures and major clonic seizures.

## Suppression of spreading seizures by focal inhibitory DREADDs

To further assess the efficacy of inhibitory DREADDs in suppressing severe forms of seizure, such as focal-to-bilateral seizures, we analyzed the spatial spread of the seizure discharges (Fig. 4). During the initial phase when spikes emerged (Fig. 2a-ii), their spatial extent was limited, having a single peak profile in the amplitude map (Fig. 4a-ii) with a half-maximum area extending over 14% (9/63) of the channels (Fig. 4b-ii). When the multi-spike-wave complexes appeared (Fig. 2a-iii and Supplementary Fig. 7a), the spatial extent was still limited, having a similar spatial configuration but with a slight expansion (21% [13/63] in the half-maximum map) (Fig. 4ab-iii). However, during status epilepticus, when sustained multi-spike-complex appeared (Fig. 2a-iv), the spatial extent of the discharges spread to the entire recording region (63/63; covering 324 mm$^2$) of the target hemisphere (Fig. 4a, b-iv and Supplementary Fig. 7b). In monkey #2, who had ECoGs implanted in both hemispheres, seizures in similar conditions spread from the induced area to the opposite hemisphere (Supplementary Fig. 8b–e, -ii to -v). Concurrently, clonic seizures also spread to body parts governed by the non-DREADD hemisphere (e.g., left hand; Fig. 2b and Supplementary Fig. 8g). Remarkably, the anti-seizure effect of DCZ was not limited

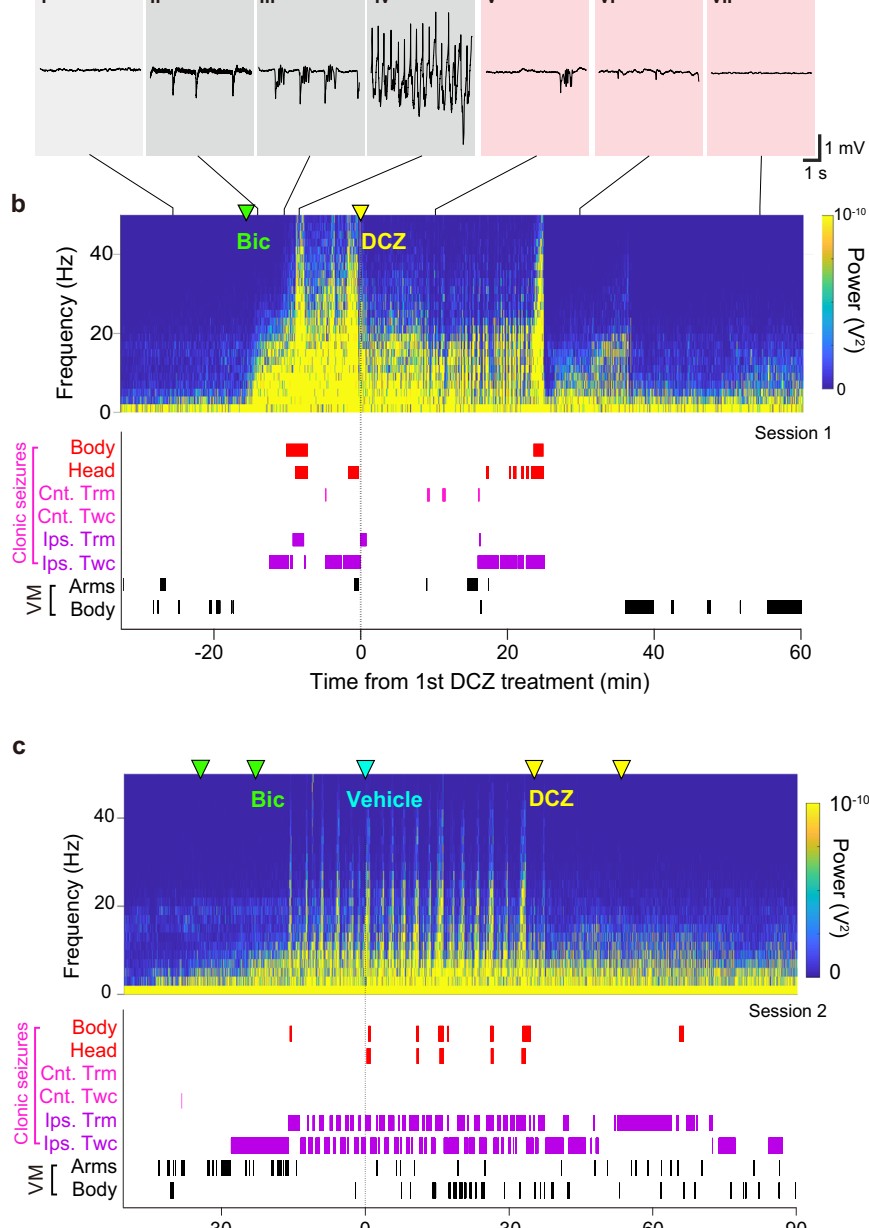

**Fig. 2 | Chemogenetic suppression of bicuculline-induced seizure and clonic seizures. a, b** Example of a DCZ-treatment session. **a** Excerpts of raw ECoG showing typical waveforms before and after bicuculline infusion and DCZ administration. Typical waveforms are depicted for baseline (i), spikes (ii, vi), multi-spike-wave complexes (iii, v), a sustained multi-spike complex (status epilepticus, iv), and the return to baseline (vii). **b** (top) Spectrogram showing seizure activity recorded from

the same electrode contact shown as a green arrowhead in Fig. 1f. (bottom) Clonic seizures and voluntary movements (VM). Ticks indicate time points of clonic seizures in the body (Body) and head (Head), contralateral hand/arm tremor (Cnt. Trm), contralateral twitch (Cnt. Twc), ipsilateral hand/arm tremor (Ips. Trm), and ipsilateral twitch (Ips. Twc). See Methods for detailed definitions. **c** Example of a vehicle-treatment session. Data were from monkey #1.

to the DREADD-expressing region but extended to the entire recording region after the seizure had spread (Fig. 4ab-v to -vii). Specifically, DREADD/DCZ-induced inactivation in the seizure focus was sufficient to reverse seizures even after a focal-to-bilateral tonic-clonic seizure has occurred (e.g., Supplementary Fig. 8b–e, -vi to iix; all four cases in monkey #2, see Supplementary Table 2).

**Histological assessment of biological reactions in the target region**

To assess the safety of gene transfer and the effect of repetitive DREADD activation in the targeted cortical region, we histologically examined neuronal loss and immune responses in the target area

(Fig. 5). We found a slight loss of layer five giant pyramidal neurons in the drug-injected part of the ipsilateral cortex (Fig. 5b-ii, c-iv), which is a known phenomenon in epileptic motor cortex[22,23]. However, in the DREADD-expressing regions distal from the bicuculline infusion site, we found no evidence of neuronal loss (Fig. 5c-v) nor any spatial heterogeneity of reporter protein expression (GFP co-expressed with hM4Di) (Fig. 5a-i, c-ii). In contrast to the homogeneous GFP expression (Fig. 5c), prominent immune responses and inflammation were localized to the seizure-induced site, as visualized by immunohistochemical staining for a cluster of differentiation 8 (CD8, cytotoxic T-cell marker) and ionized calcium-binding adapter molecule (Iba1, microglial cell marker) (Fig. 5d–f). Similarly,

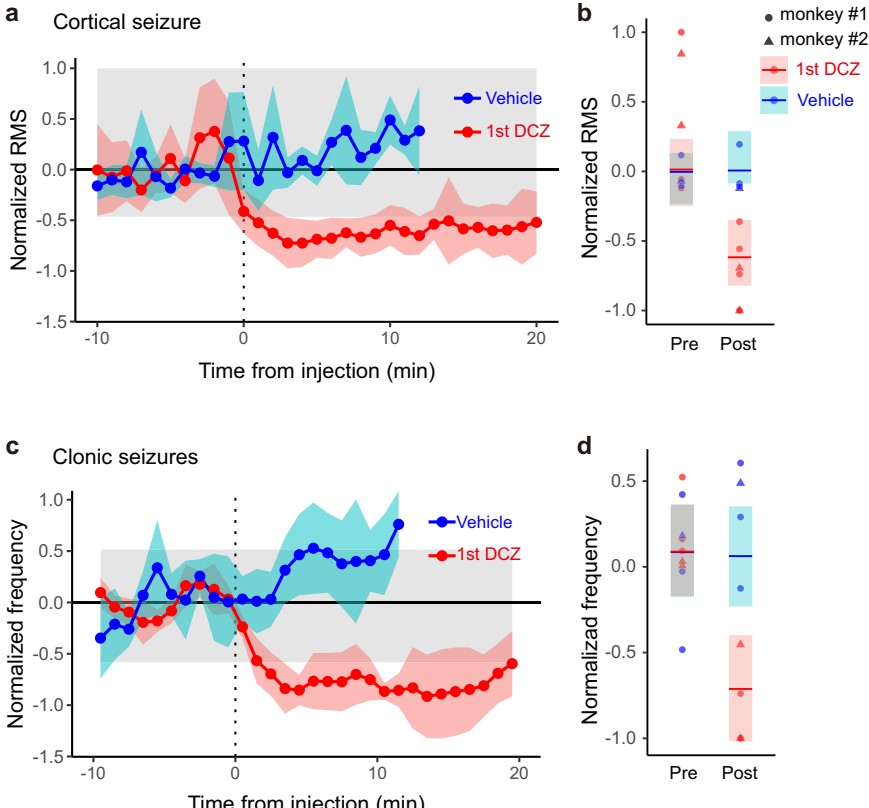

**Fig. 3 | Rapid effect of DCZ application on cortical and clonic seizures.**
**a** Changes in normalized seizure amplitude over time after treatment. Connected dots indicate seizure amplitude defined as the root-mean-square (RMS) of the ECoG signal in the hemisphere injected with bicuculline normalized by changes in average RMS during the pretreatment period (−10 to 0 min before treatment) and scaled by the maximum values in individual DCZ sessions and vehicle sessions (see Methods and Supplementary Table 1). Shaded regions around the dots indicate the standard error. The gray shaded area represents the 95th percentile of the values during the pretreatment period. **b** Evaluation of treatment effect on seizure amplitude using a Bayesian space-state model that considers subject-related effects. The maximum a-posteriori (MAP) estimators (points and bars for a single session and the mean, respectively) and their 99% credible intervals (shaded

rectangles) for seizure amplitudes 3 min before and after administering DCZ and vehicle are plotted; the post-treatment intervals do not overlap, indicating that DCZ significantly triggered a drop in amplitude. n = 2 animals examined over 6 (DCZ) and 4 (Vehicle) independent experiments. **c** Changes in normalized frequency of clonic seizures (mean ± standard error) over time observed in the hand/arm/torso/face after DCZ (red) and vehicle treatment (blue) (see Supplementary Fig. 4 for raw data). **d** The MAP estimators and their 99% credible intervals for clonic seizures 3 min before and after administering DCZ and vehicle. The two treatments differentially affected the frequency of clonic seizures during pre- and post-treatment periods (the 99% credible post-treatment intervals do not overlap). n = 2 animals examined over 5 (DCZ) and 4 (Vehicle) independent experiments.

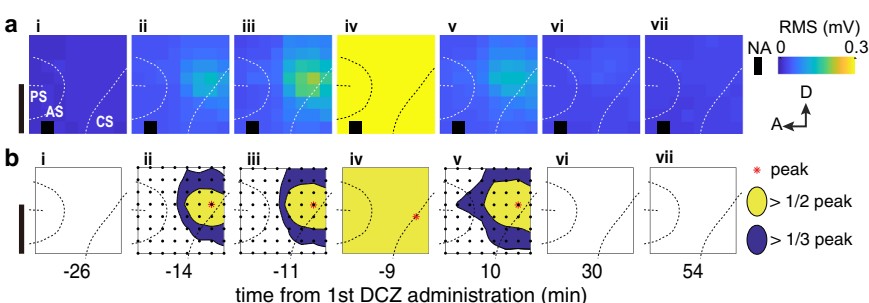

**Fig. 4 | Chemogenetic attenuation of wide-spread seizures. a** Example seizure amplitude maps are shown as the root-mean-square (RMS) of the raw signals in a pseudo-color scale. **b** Half and one-third peak maps showing the extent of the signal spread, normalized by the peak size. **i** through **vii** correspond to the time points shown in Fig. 2a. Data were from monkey #1. NA non-available recording channel. Scale bars: **a**, **b** 10 mm.

immunohistological observations from monkey #2, who had DREADD expression in the left hemisphere, and ECoG electrode implantation and bicuculline injections in both hemispheres (Table 1) showed no clear immune response-related signal caused by DREADD expression. Moderate neuronal loss was observed in the right MI (Supplementary Fig. 9a) and was not colocalized with the

GFP-positive regions (Supplementary Fig. 9a, b, e, f). Prominent immune responses were found along the needle tracks as visualized by a strong Glial Fibrillary Acidic Protein (GFAP, astrocyte marker) signal (Supplementary Fig. 9c, g). Collectively, these results suggest that hM4Di delivery, transduction, and activation did not cause any undue tissue damage.

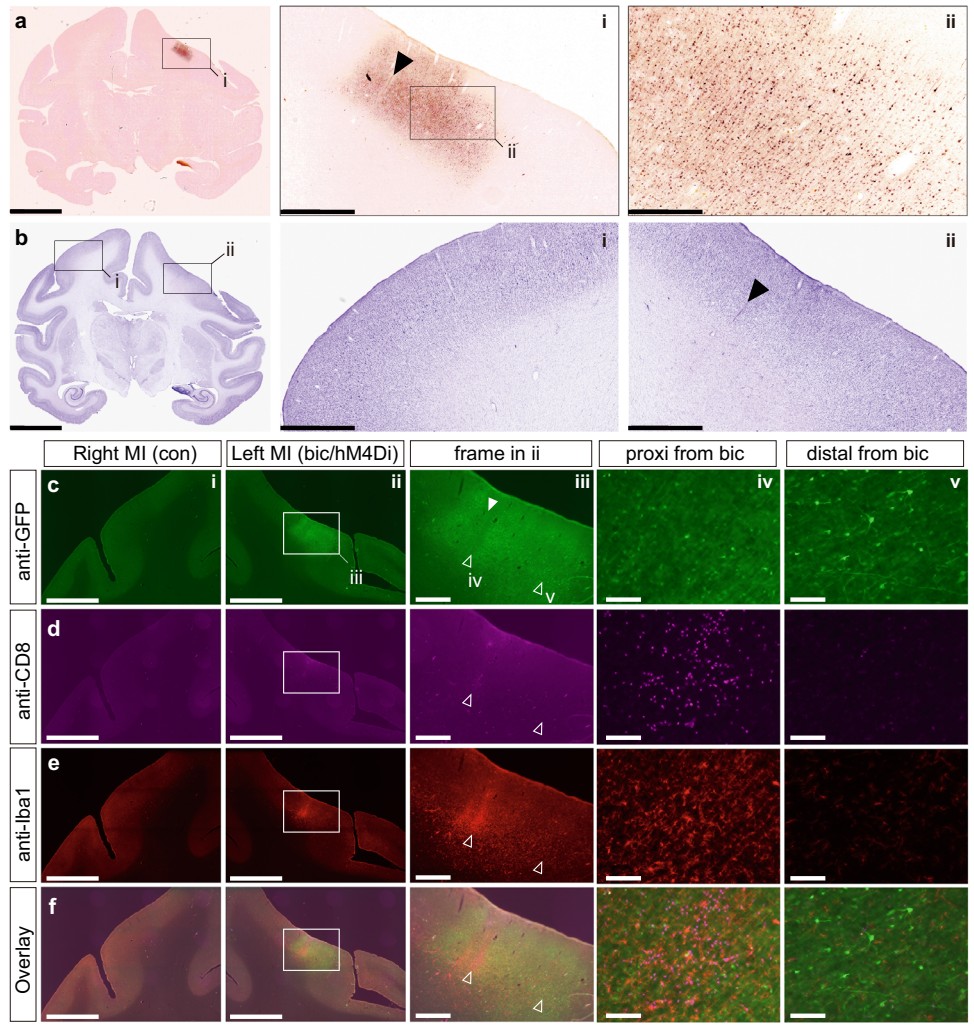

**Fig. 5 | Histological confirmation in monkey #1. a** Anti-GFP immunohistochemical staining. Frames in **a** and **a-i** depict the respective enlarged areas shown in **a-i** and **a-ii**, respectively. Images were digitally enhanced by redefining the tone curves linearly. **b** Nissl staining. The left (right) frame in **b** depicts the respective enlarged areas shown in **b-i** and **b-ii**. **c** Anti-GFP immunofluorescent staining. **d** Anti-CD8 immunofluorescent staining. **e** Anti-Iba1 immunofluorescent staining. **f** Overlay images. Closed arrowheads point to the bicuculline infusion site. Open arrowheads point to the high-magnification images proximal (left: **c-iv** through **f-iv**) and distal (right: **c-v** through **f-v**) from the bicuculline infusion site. Scale bars: **a**, **b**, 10 mm; **c-i** through **f-i**, **c-ii** through **f-ii**, 5 mm; **a-i**, **b-i**, **b-ii**, **c-iii** through **f-iii**, 1 mm; **a-ii** 250 μm; **c-iv** through **f-iv**, **c-v** through **f-v**, 100 μm. Similar observations were made on monkey #2 (See Supplementary Fig. 9).

## Discussion

Here we demonstrated that activating hM4Di located in the seizure focus is sufficient for suppressing cortical seizures in an NHP model of frontal lobe epilepsy. Activation of hM4Di effectively attenuated or shut down seizures even after they had become quite severe, such as reaching status epilepticus or a spatially wide-spread state. To the best of our knowledge, this is the first successful proof-of-concept demonstration of chemogenetic seizure suppression in NHP, representing an advance toward clinical therapeutic applications.

Previous preclinical studies in rodents have shown that inhibitory DREADDs effectively suppress cortical seizures when they are focally transduced into excitatory neurons[15,16,18], as do excitatory DREADDs when transduced into inhibitory neurons[16]. Other studies have reported successful suppression of focal-to-bilateral seizure by transducing inhibitory DREADDs into the thalamus[17]. The present study extends these preclinical findings by demonstrating chemogenetic seizure suppression in NHPs. By taking advantage of the closer similarity in brain size, complexity, and genetic background that NHPs have with humans[19,20], this study may provide a reference for future human clinical applications.

The first critical step for successful chemogenetic manipulation is sufficient gene delivery to the target region and stable expression, which directly relates to the efficacy of chemogenetic manipulation[24,25]. The difference between rodents and NHPs has been pointed out in the choice of optimal viral vectors, injection parameters, and optogenetic evaluation methods, which also require highly efficient gene delivery[26]. In the current study, multi-track injections with the mosaic AAV vector[27] resulted in hM4Di expression covering an approximately 3 × 3-mm cortical region, which was validated by immunohistochemical analysis. In addition, we used DREADD-PET imaging to assess hM4Di expression in vivo before DREADD activation, showing increased specific tracer binding ($\Delta BP_{ND} > 0.5$) that roughly matches the value we observed in our previous studies[12,28–30]. Presumably, PET imaging can also be used with human patients, allowing for long-term monitoring of DREADD expression. Lastly, we confirmed the biological safety of the DREADD system with respect to brain tissue, as inflammation overlapped with the site of repetitive seizure induction, but not with the DREADD-expressing region.

The second critical step for successful chemogenetic manipulation is the delivery of a chemogenetic agonist. The first-generation

DREADD actuator, CNO, has very poor permeability in the brain, and a certain amount of injected CNO is reverse-metabolized to clozapine, which raises a concern about potential off-target effects[13,25]. Our data indicate that systemic administration of DCZ is effective for suppressing cortical and clonic seizures within 1 and 3 min, respectively. The rapid chemogenetic action of DCZ is consistent with previous reports and will be extremely useful in prospective therapeutic applications. Here, we used the same DCZ dose (0.1 mg kg$^{-1}$) as in previous reports[12,30,31], which significantly blocked output from the DREADD-expressed motor cortex. Importantly, also as in previous reports[12,31–33], we did not observe any discernible side effects on brain activity or animal behavior. In the current study, we administered DCZ intramuscularly, but it can also be administered orally and has been proven to activate hM4Di chronically for more than two weeks with repetitive doses[34]. Although further safety testing is needed before moving on to clinical trials, these characteristics suggest that DCZ may be a safe and effective DREADD actuator that might be used in future clinical trials.

While the anti-convulsant effects of DCZ were roughly consistent throughout the 1-h session, the anti-seizure effect in the MI of monkey #2 only lasted about 20 min. This is not likely related to a rapid decrease in DCZ because the second dose of DCZ had no additional impact on either seizures or convulsions. A more plausible reason for the short-lived effect is that monkey #1 and monkey #2 differed in some critical way. Indeed, the PET signal in monkey #2 was lower than that in monkey #1 (Supplementary Fig. 1a), indicating relatively weak hM4Di expression in this monkey. This was further reflected in GFP immunohistochemistry in monkey #2, indicating that hM4Di expression in the superficial layer of MI was much lower than in the deep layer. In contrast, hM4Di expression in the superficial and deep layers of monkey #1 was equally dense (Supplementary Fig. 1). The low expression in the superficial layers of monkey #2 might have led to the return of cortical seizures in the later sessions when bicuculline could have diffused to the superficial layers. This view is consistent with results for clonic seizures; because the deep layers of MI in both monkeys expressed enough hM4Di, its output was suppressed by DCZ administration, which led to continuously attenuated convulsive behavior for 1 h.

The present study employed an NHP model of FLE with the focus located within the primary motor cortex. In such cases, partial resection is often carried out as a compromise between the risk of motor deficits and the prevention of seizures[3–6]. Thus, FLE that is focused in the motor area is a potential target for on-demand chemogenetic therapy. The severity of focal FLE increases when seizures spread widely from one hemisphere to another, which eventually causes loss of consciousness, and/or strong clonic seizures that could result in serious injuries and accidents. We observed spatially wide-spread seizures characterized by continuous refractory waveforms that spread over the entire recorded area and were accompanied by clonic seizures in the body. These spatially wide-spread seizures, and even the bilaterally spread seizures, were largely attenuated by activation of inhibitory DREADDs around the original epileptic focus, probably because the surrounding inhibitory network (not directly affected by bicuculine) began to function normally. We also verified that the dosage of the DREADD actuator drug did not significantly affect general motor or cognitive ability. These results suggest that the chemogenetic system may be useful in therapeutic settings; patients who have had an inhibitory DREADD transduced into their epileptic foci could take the actuator drug on-demand to prevent secondary generalization of seizures with minimal nonspecific side effects.

The present study has several major limitations that should be overcome in future studies. First, we adopted an NHP model with acute pharmacological seizure induction, which is useful for testing the efficacy of the DREADD system repetitively in focal or focal-to-bilateral forms of seizures. However, with this model, we cannot persuade its long-term efficacy in reducing chronic recurring wide-spread seizures of epilepsies. For this, the DREADD system needs to be tested in chronic models of epilepsy, such as a kindling model[35], in which the mechanisms of seizure propagation are more complex. Second, the current study was performed under xylazine sedation because bodily convulsions had to be clearly distinguished from voluntary movements. Future studies are needed to directly determine the extent to which convulsions are suppressed in non-sedated animals, excluding a potential synergistic effect. Third, the present study demonstrated chemogenetic seizure attenuation but not its complete cessation, indicating that there is room for improvement. One possibility is the choice of the promoter to drive expression. We used the *hSyn* promoter to drive neuron-specific expression of the inhibitory DREADD. However, targeting excitatory neurons exclusively will be more effective; a rodent study has demonstrated the effectiveness of using the *CaMKIIa* promoter that allows selective expression in excitatory neurons[16]. Finally, the present report is a proof-of-concept study demonstrating that hM4Di expressed in cortical neurons sufficiently suppresses the spread of cortical and clonic seizures whose foci are in the motor area. However, in an actual clinical setting, the epileptic focus needs to be localized before introducing the DREADDs. The accuracy of focus localization has been largely improved by combining imaging/electrophysiological methods (e.g., MRI, FDG-PET, MEG, EEG, and ECoG) with current algorithms[36,37]. These points need further confirmation in future studies using NHP models of epilepsy.

In summary, the present study demonstrated for the first time in NHPs that inhibitory DREADD hM4Di combined with its actuator DCZ is capable of attenuating cortical seizures. A combination of PET-guided chemogenetics and multimodal seizure detection in an NHP model will fill in the physical and genetic gaps in knowledge between the effective chemogenetic strategies that have been proven in rodent models and what can be achieved in primates. This approach may be crucial for simulating the therapeutic process used in future human clinical settings.

## Methods

### Subjects

All experimental procedures involving animals were carried out in accordance with the Guide for the Care and Use of Nonhuman primates in Neuroscience Research (The Japan Neuroscience Society; https://www.jnss.org/en/animal_primates) and were approved by the Animal Ethics Committee of the National Institutes for Quantum Science and Technology. Two male cynomolgus macaque monkeys (*Macaca fascicularis*; 5 years old; 4.8 kg, 6 years old; 5.4 kg) were used for the experiments. The animals were provided by HAMRI Co., Ltd., Japan. A standard monkey chow diet, supplementary fruits/vegetables, and a vitamin C tablet (200 mg) were provided daily.

### Viral vector production

The AAV2.1-hSyn-FLAG-hM4Di-IRES-AcGFP vector was produced by helper-free triple transfection and purified using affinity chromatography (GE Healthcare, Chicago, USA). Viral titer was determined by quantitative PCR using Taq-Man technology (Life Technologies, Waltham, USA). For the production of the AAV2.1 vector, the pAAV-RC1 plasmid-coding AAV1 capsid protein and the pAAV-RC2 plasmid-coding AAV2 capsid protein were transfected in a 1:9 ratio[27].

### Surgical procedures

Surgeries were performed under aseptic conditions in a fully equipped operating suite. We monitored body temperature, heart rate, SpO$_2$, and tidal CO$_2$ throughout all surgical procedures. Anesthesia was induced using intramuscular (i.m.) injection of ketamine (5–10 mg kg$^{-1}$) and xylazine (0.2–0.5 mg kg$^{-1}$). Monkeys were intubated with an endotracheal tube and anesthesia was maintained with isoflurane (1–3%, to effect). After surgery, prophylactic antibiotics and analgesics were administered. Before the surgeries, magnetic resonance (MR)

imaging (a preclinical 7 Tesla 40-cm bore MRI system, BioSpec, Avance NEO system, Bruker Biospin, Ettlingen, Germany) and X-ray computed tomography (CT) scans (Accuitomo170, J. MORITA CO., Kyoto, Japan) were acquired under anesthesia (continuous infusion of propofol 0.2–0.6 mg kg$^{-1}$ min$^{-1}$, i.v.). Overlaid MR and CT images were created using PMOD® image analysis software (PMOD Technologies Ltd, Zurich, Switzerland) to estimate the stereotaxic coordinates of target brain structures.

## AAV vector injection

Monkeys #1 and #2 were injected with AAV2.1-hSyn-FLAG-hM4Di-IRES-AcGFP ($3.5 \times 10^{12}$ and $3.5 \times 10^{13}$ particles per mL, respectively). The injections were performed under direct vision. The putative hand/arm target region of the left MI was determined based on sulcal landmarks from a pre-scanned MRI (Fig. 1b)[38]. After retracing the skin and galea, the cortex was exposed by removing a bone flap and reflecting the dura matter. Vectors were pressure-injected using a 10-μL syringe with a 33-gauge needle (NanoFil Syringe, WPI, Sarasota, USA). The syringe was mounted on a motorized microinjector (UMP3T-2, WPI) that was attached to a manipulator (Model 1460, David Kopf, Ltd., Tujunga, USA) on the stereotaxic frame. The injection needle was then inserted into the brain and moved down to 3.0 mm below the surface. After 5 min, the needle was pulled up 0.5 mm, and 1.5 μL per site of the vector solution was injected at 0.15 μL min$^{-1}$. The needle was slowly pulled up following an additional 5-min waiting period to prevent backflow. In total, virus vector injections were conducted at nine sites for each monkey, with an inter-site distance of ~1.2 to 1.5 mm along the cortical surface (total 13.5 μL) into the target region (Fig. 1b). The dura matter was stitched back with absorbable suture thread (VICRYL Plus, Johnson & Johnson, New Brunswick, USA). The bone flap was also placed back and fixed to the skull with absorbable thread via drilled cranial holes before closing the galea and the skin.

## PET imaging

To examine the expression of hM4Di in vivo, PET imaging was conducted before vector injection and approximately six weeks after the virus-injection surgery, as previously reported[12]. Briefly, the monkeys were anesthetized with intramuscular injection of ketamine (8 mg kg$^{-1}$) and xylazine (0.4 mg kg$^{-1}$), along with isoflurane (1–3%) during all PET procedures. PET scans were performed using a microPET Focus 220 scanner (Siemens Medical Solutions USA, Malvern, USA). Transmission scans were performed for approximately 20 min with a Ge-68 source. Emission scans were acquired in 3D list mode with an energy window of 350–750 keV after intravenous bolus injection of [$^{11}$C]DCZ (147-364 MBq). The actual injected dose depended on the specific radioactivity and the weight of the monkeys. Emission-data acquisition lasted 90 min. PET images were reconstructed with filtered back-projection using a Hanning filter cut-off at a Nyquist frequency (0.5 mm$^{-1}$). To estimate the specific binding of [$^{11}$C]DCZ, the regional binding potential relative to non-displaceable radioligand (BP$_{ND}$) was calculated using PMOD with an original multilinear reference-tissue model (MRTMo) and the cerebellum as a reference[12,39]. The expression of transduced DREADD in vivo was confirmed in contrast images (subtracted from PET images taken before vector injection) using SPM12 software (https://www.fil.ion.ucl.ac.uk/spm/software/spm12/) running in a MATLAB R2016a (The Mathworks, Natick, USA) environment. Contrast PET images were registered to individual structural MR images using PMOD. 3D-rendered multimodal overlay images were built using a medical image analysis workstation (AZE Virtual Place 350, Canon Medical Systems, Ohtawara, Japan).

## ECoG and other device implantation

Sixty-four channel ECoG electrodes were prepared with a micromachine using 0.25-μm-thick gold wiring and 10-μm-thick Parylene-C insulation with the recording contacts exposed in a 100 × 100-μm-square shape[27,40–42]. Contacts were configured in an 8 × 8 grid shape with 2.5-mm spacing. The lead wires and Parylene-C insulation were aligned in columns separated by slits (Fig. 1f and Supplementary Fig. 8a). A pair of cable bundles led from the electrode to two 36-pin connectors with 0.025-inch pitch (Supplementary Fig. 8a; #A8828-001-vv, Omnetics, MN, USA)[40]. We implanted one ECoG electrode in monkey #1 (left MI) and two in monkey #2 (bilateral MI). The bone flaps were removed again above the vector injection loci to make cranial windows that were ~24 mm (antero-posterior) × 20 (mediolateral) mm in area (Fig. 1f and Supplementary Fig. 8a). An ECoG film electrode was placed epidurally within each cranial window with the edge of the ECoG electrode and the artificial dura inserted beneath the skull for stability (Fig. 1f). A conventional box-shaped recording chamber was placed above the ECoG electrode in the hM4Di-expressing hemisphere (left in both animals). The chamber, housing case(s) for the ECoG connector(s), and a headpost were fixed to the skull with titanium anchor screws and dental acrylic during the same surgery. Wires were also implanted in the bilateral forearms (brachioradialis muscles) of monkey #2 for EMG recording (Supplementary Fig. 8f, g).

## Epileptogenic drug administration

To determine the location for epileptogenic drug infusion, we performed CT scans before the day of drug delivery while a dummy cannula was placed above the dura, within the slit of the ECoG electrode, and above the dura at the site of AAV injection. CT, MR, and PET images were aligned with PMOD® software using the "Fusion" mode to confirm that the cannula was positioned directly above the hM4Di-expressing region (Fig. 1f). On the day of the epilepsy-induction experiment, animals were placed in a monkey chair with their heads fixed. To monitor and evaluate convulsive body movements and to distinguish them from voluntary movements, monkeys were sedated with intramuscular injection of xylazine (0.8–1.2 mg kg$^{-1}$ initial dose, and 0.4–0.8 mg kg$^{-1}$ h$^{-1}$ for maintenance). Care was taken not to make the injections within 10 min before/after vehicle or DCZ treatment to minimize the effect of xylazine on treatment-induced changes in seizure/convulsion. A syringe with a 33-gauge needle (NanoFil Syringe, WPI) was filled with bicuculline (4 μg μL$^{-1}$ in PBS) that was freshly prepared on the day of drug delivery. The syringe was mounted to a motorized microinjector (Mode Legato 160, KD Scientific, Holliston, USA) on a manipulator (Model 1460, David Kopf, Ltd.) connected to a customized arm that was attached to the monkey chair. The injection needle was pierced directly through the dura and inserted into the brain to a depth of ~4.0 mm below the dura surface. After 5 min, the needle was moved 1.5 mm back and 0.4–16 μg (0.1–4 μL) of bicuculline was pressure-injected at 1 μL min$^{-1}$ over 1–2 injection trials until epileptic twitches were observed in the contralateral hand/arm (Supplementary Table 1). Bicuculline was delivered 10–15 min after baseline monitoring of ECoG and body movements. The microinjector was connected to the controller only during the drug injection period to reduce electrical humming noise in the electrophysiological recordings. Each recording session terminated 1–2 h after bicuculline delivery, followed by i.m. injection of Diazepam (0.2–0.8 mg kg$^{-1}$)—a long-lasting anti-epileptic drug for monkeys—before returning them to the home cage. Monkeys were visually inspected for abnormal behavior and were treated with additional doses of Diazepam (0.1–0.4 mg kg$^{-1}$) for two more days. The ECoGs and video data from 15 min before bicuculline injection until 60 min after DCZ administration were used for the analyses; data after Diazepam injection was not included.

## Chemogenetic drug administration

DCZ (HY-42110, MedChemExpress) was dissolved in dimethyl sulfoxide (DMSO, FUJIFILM Wako Pure Chemical Co.), then diluted in saline to a final volume of 1 mL (2% DMSO in saline), thus achieving 0.1 mg kg$^{-1}$ dose. Note that this dose of DCZ yields 80–90% hM4Di occupancy and affects behavioral performance via hM4Di activation in

monkeys[12]. Vehicle control was an i.m. injection of vehicle solution (2% DMSO in saline) at the same volume. Vehicle was delivered 15–20 min after delivery of the bicuculline. The first injection of DCZ was delivered 15–20 min after bicuculline (one case) or vehicle (5 cases). The second injection of DCZ was delivered 18–30 min after the first one (five cases). The timing of vehicle and DCZ administration for each session is summarized in Supplementary Table 1.

## Behavioral examination

Manual dexterity of a monkey (monkey #2) expressing hM4Di in the putative hand region of MI was tested using a modified Brinkman board task[43,44]. A plastic board with 20 rectangular slots (5 × 10 mm with a 5-mm depth) was fixed in front of a monkey chair facing the animal. The monkey picked up a small piece of sweet potato (~4 mm in diameter) from each slot for immediate consumption. The monkey consistently used its thumb and index finger to pick up the pellets (i.e., precision grip)[31,45]. In each trial, the monkey was allowed to operate for 25 s with one of its hands (contralateral or ipsilateral to the hM4Di-expressing hemisphere), and the number of pellets successfully picked and eaten was counted. In a single experimental session, the monkey performed 8 trials: two untreated trials for each hand, and two more trials after each i.m. administration of vehicle or DCZ. When following the treatments, the tests began 15 min after injection. The tested hand alternated each trial, and the initial hand was randomized in each experimental session.

## Data acquisition

**ECoG**. ECoG data were recorded using a TDT System3 combined with Intan headstages. Signals were fed to headstage-amplifier/digitizers (RHD2132, Intan Technologies, Los Angeles, USA) and a preprocessing unit (PZ2, TDT), then fed into the digital signal-processing module (RZ2, TDT). The ECoG signal was band-pass filtered between 1.5 and 500 Hz digitally and stored at 3 kHz. Data were acquired using Open developer software (TDT, version 2.16) and analyzed with in-house programs running on the MATLAB R2016a environment[42]. Signals were digitally processed with a high-pass filter (1 Hz, tenth order) and a band-stop filter (48–52 Hz, 20th order). The root-mean-square (RMS) of the waveform and its temporal average were computed as the amplitude of the seizure. Spectral power was computed using the short-time Fourier transform and normalized for each frequency using the baseline power averaged within a 3-min time window 15 to 12 min before bicuculline injection. Data were visualized either with MATLAB or GraphPad Prism 9 (GraphPad Software, San Diego, USA).

**Video**. Animal behavior was monitored in the head/chest region and the torso/arm/leg region using two video cameras (MTC-9272, Mother Tool, Ueda, Japan; WAT-232S, Watec, Tsuruoka, Japan). Videos were stored in a multi-channel video recorder (RD4304, ARUCOM, Fukuoka, Japan). Stored movies were viewed offline with video software (ifileplaypack, ARUCOM). We noted the following convulsive or voluntary movements, and their timing and frequency were analyzed by an observer familiar with monkey behavior: "Twitch" (a sudden jerk of the hand/arm or leg), "Tremor" (continuous and rhythmical shaking of the hand/arm that lasted for longer than 2 s), clonic seizure in the "Body" (large scale shaking of the body, including torso and legs), clonic seizure in the "Head" (whole-face movements ranging from small twitches in the unilateral lip and/or eyebrows to large scale jerking), "Voluntary movements" (non-periodical large movement of the arms and/or body).

## Histology

For histological inspection, monkeys were immobilized with ketamine (10 mg kg$^{-1}$, i.m.), deeply anesthetized with an overdose of sodium pentobarbital (80 mg kg$^{-1}$, i.v.), and then transcardially perfused with saline at 4 °C, followed by 4% paraformaldehyde in 0.1 M phosphate-

buffered saline (PBS) at a pH of 7.4. The brains were removed from the skull, postfixed in the same fresh fixative overnight, saturated with 30% sucrose in phosphate buffer (PB) at 4 °C, and then cut serially into 50-μm-thick sections on a freezing microtome. For visualization of AcGFP immunoreactive signals, each sixth section was immersed in 1% skim milk for 1 h at room temperature and incubated overnight at 4 °C with rabbit anti-GFP monoclonal antibody (1:500; G10362, Thermo Fisher Scientific, MA, USA) in PBS containing 0.1% Triton X-100 and 1% normal goat serum for 2 days at 4 °C. The sections were then incubated in the same fresh medium containing biotinylated goat anti-rabbit IgG antibody (1:1,000; Jackson ImmunoResearch, West Grove, PA, USA) for 2 h at room temperature, followed by avidin-biotin-peroxidase complex (ABC Elite, Vector Laboratories, Burlingame, CA, USA) for 1.5 h at room temperature. For visualization of the antigen, the sections were reacted in 0.05 M Tris-HCl buffer (pH 7.6) containing 0.04% diamino-benzidine (DAB), 0.04% NiCl$_2$, and 0.003% H$_2$O$_2$. The sections were mounted on gelatin-coated glass slides, air-dried, and cover-slipped.

Additional triple immunofluorescence immunohistochemistry for AcGFP, CD8, and Iba1(Fig. 5) and for AcGFP, NeuN, and GFAP (Supplementary Fig. 9) was conducted as reported previously[46]. Briefly, sections were pretreated as described above, and then incubated with goat polyclonal anti-GFP antibody (1:500 dilution; OriGene, USA), rabbit monoclonal anti-Iba1 antibody (1:1000 dilution; Wako), mouse monoclonal anti-CD8 antibody (1:1000 dilution; Bio-Rad, USA), anti-NeuN antibody (1:2000 dilution; Millipore, USA), and anti-GFAP antibody (1:1000 dilution; Sigma, USA) for 2 days at 4 °C. The sections were subsequently incubated for 2 h at room temperature with a cocktail of Alexa 488-conjugated donkey anti-goat IgG antibody (1:400 dilution; Invitrogen), Alexa 555-conjugated donkey anti-rabbit IgG antibody (1:200 dilution; Invitrogen), and Alexa 647-conjugated donkey anti-mouse IgG antibody (1:200 dilution; Invitrogen). Images of histo-chemical sections were captured using a digital slide scanner (Nano-Zoomer S60, Hamamatsu Photonics K.K., Hamamatsu, Japan; 20× objective, 0.46 μm per pixel). The histochemical sections were scanned as bright-field images and exported as 8-bit TIFF images using viewer software (NDP.view2, Hamamatsu Photonics K.K.). Immuno-fluorescent immunohistochemical images were captured using a microscope equipped with a high-grade charge-coupled device (CCD) camera (Biorevo, Keyence, Japan) and exported as 8-bit TIFF images.

## Statistical analyses

**ECoG, video, and behavioral experiment analyses**. All data were preprocessed using custom programs written in MATLAB and were analyzed using GraphPad Prism 9 (GraphPad Software, San Diego, USA) or the R statistical computing environment, version 4.0.3 (R Development Core Team, Vienna, Austria). To quantify the effects of DCZ and vehicle treatment on cortical seizure, the amplitude was first calculated as the RMS of the waveform in each 1 min bin before (−15 to 0 min) and after treatment (0 to 60 min and 0 to 15 min for DCZ and vehicle, respectively). Then the values were normalized by the average value during each session's pre-administration period. We considered a change to be significant if the measured value deviated from the 95th percentile of the values during the pre-administration period. For display purposes, the results were then scaled by the maximum absolute values for the pre- and post-administration periods (−15 to 12 min). For clonic seizures, the frequency of convulsions was calculated for each 1 min bin and smoothed with 5-min windows shifted by 1 min. The other procedures were the same as those for cortical seizures. To further examine the impact of treatments on cortical and clonic seizures, while taking into account the effect of the subject, we constructed a Bayesian state-space model characterizing the trend of the normalized values during the pre- and post-administration periods by means of R packages (CmdStan ver. 2.30.1 and cmdstanr ver. 0.5.2). The observation model was defined as the sum of three trends and

measurement error ε as follows,

$$y = \hat{y} + u d_M + v d_T + \varepsilon, \quad \varepsilon \sim \mathrm{Normal}(0, \sigma^2), \tag{1}$$

where $y$ denotes the normalized value, $\hat{y}$ denotes the basic trend, $d_M$ denotes the random effects by subject, and $d_T$ denotes the effects of treatment. Parameters were weighted as follows:

$$u = \begin{cases} -1 \,(\text{monkey\#1}) \\ 1 \,(\text{monkey\#2}) \end{cases}, \quad v = \begin{cases} 0 \,(\text{vehicle}) \\ 1 \,(\text{DCZ}) \end{cases}. \tag{2}$$

Random effects by subject were sampled by the normal distribution

$$d_M \sim \mathrm{Normal}(0, \sigma_M^2), \tag{3}$$

and for $d$ and $d_T$, process models were defined as follows,

$$\hat{y}_{(t)} - \hat{y}_{(t-1)} \sim \mathrm{Normal}\left(\hat{y}_{(t-1)} - \hat{y}_{(t-2)}, \sigma_{\hat{y}}^2\right), \tag{4}$$

$$d_{T(t)} \sim \mathrm{Caushy}\left(d_{T(t-1)}, \sigma_T^2\right). \tag{5}$$

For evaluation, we compared the means of the maximum a-posterior (MAP) estimators and their 99% credible intervals for the basic trend with random effects by subjects, $\hat{y} + v d_T$, 3 min before and after administering treatments (DCZ or vehicle) (Fig. 3b, d).

### Reporting summary

Further information on research design is available in the Nature Portfolio Reporting Summary linked to this article.

## Data availability

The data that support the findings of this study are available from the authors upon request. The source data generated in this study have been deposited in the Open Science Framework database (http://github.com/minamimoto-lab/2022-Miyakawa-epilepsy). Source data are provided in this manuscript. Source data are provided with this paper.

## Code availability

The custom MATLAB and R scripts are available from the authors upon request.

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

## Acknowledgements

We thank Tomomi Kokufuta, Rie Yoshida, Jun Kamei, Ryuji Yamaguchi, Yoshio Sugii, Yuichi Matsuda, Takashi Okauchi, Maki Fujiwara, Mayuko Nakano, Emiko Nakano, Sayaka Shibata, Nobuhiro Nitta, and Maiko Ono for their technical assistance. We also thank Dr. Tomohiko Takei for providing expert knowledge regarding EMG surgery and Katsushi Kumata and Dr. Ming-Rong Zhang for producing the radioligand. This study was supported by QST President's Strategic Grant (Creative Research) (to N.M.), by MEXT/JSPS KAKENHI Grant Numbers JP19K07811, JP20H04596 (to N.M.), and JP19K08138 (to Y.N.), and by AMED Grant Numbers JP18dm0307007 (to T.H.), JP21dm0207077 (to M.T.), and JP16dm0107146 (to T. Minamimoto).

## Author contributions

N.M. and T. Minamimoto conceptualized, designed, and supervised the project. N.M. and Y.N. performed the surgery. N.M. performed and analyzed the chemogenetic experiments. N.M. and Y.N. performed and analyzed the PET experiments. N.M. and K.M. performed statistical analyses. Y.H. and K.I. performed and analyzed histological experiments. N.M. and T. Minamimoto wrote the manuscript with feedback and revisions from all the coauthors.

## Competing interests

The authors declare no competing interests.
