## [Peer Review File · Nature Communications]

Peer review comments,

First round review:

Reviewer #1 (Remarks to the Author):

This manuscript by Miyakawa et al reports proof of concept experiments showing that DREADDs can be used to attenuate focal epilepsy in nonhuman primates. This is an excellent and very well written manuscript that describes well-designed studies using very sophisticated and state of the art methods and procedures. I only have minor comments.

The effect of DCZ and hM4Di activation is affecting widespread seizure activity is impressive. What do the authors think is driving this effect? Perhaps they can comment on this a bit more in the discussion.

The short-lived efficacy (short relative to its pharmacokinetics) of DCZ in attenuating seizure activity is also interesting and I wonder if the authors would consider also elaborating on this in the discussion as well.

Line 87: please consider changing to "which can have"

Figure 1: "epilepsy" is misspelled

Line 240: consider changing to "viral vectors"

Line 302: please delete the second "the"

Reviewer #2 (Remarks to the Author):

The manuscript by Miyakawa et al. titled "Chemogenetic attenuation of cortical seizures in primates" is a proof of principal study. The main goal of this study was to demonstrate the feasibility of chemogenetic technology as an alternative treatment for frontal lobe epilepsy. Using 1-2 cynomolgus macaques, this group used a pharmacologically-induced model of epilepsy in combination with inhibitory-DREADDs, positron emission tomography, and electrocorticogram. Monkeys had hM4Di introduced into the left primary motor cortex which later served as the same location for the seizure focus by the administration of a pharmacological agent, bicuculline. The main conclusion of the study shows that hM4Di is an effective way to dampen seizure severity only if expressed within the seizure focus. The premise of this manuscript has potential for high clinical impact, but the study has some issues that should be addressed.

Although it is understandable the difficulty and limited resources involved in NHP research, the feasibility of this study would be greatly strengthened by control studies in which seizure within the M1 region is elicited, behavioral, and EEG studies are performed in NHPs that do not express hM4Di. This will be important for immunohistological analysis of cell death. In addition to a comparison of outcome measures between hM4Di-activated and hM4Di non-expressing NHPs, the study would be strengthened by a comparison of the two hM4Di-expressing NHPs. There were differences between the 2 NHPs in their AAV viral titer used, the number of seizures elicited, potential damage caused by surgery from placement of ECoG channels bilaterally versus unilaterally. Additionally, it is unclear if the two hM4Di-NHPs were used for each experiment (i.e., behavior, PET scans to examine location of DREADD expression, ECoG measurements during seizure, seizure, immunohistochemistry). For example, page 6 line 125 (Sup. Fig. 1) shows that chemogenetic function of hM4i expressed in the M1 by testing grasping was only performed in monkey #2. The use of a table would be necessary to understand the use of each animal and to determine if the information presented is valid across animals. This would also allow for the much-needed clarification of method details such as how many times a seizure was elicited in each NHP, the order of treatment (ie DCZ or vehicle).

Secondly, a discussion or additional data regarding the quick onset and offset of DCZ action is needed. It seems a bit fast for the onset of DCZ action within the brain, as observed by a decrease in seizure amplitude that occurred within a 1 minute after an intramuscular injection. Additionally,

the effect of DCZ ended after only 25 minutes when the same group has shown high binding within the putamen 90min after injection and similar concentrations of DCZ within the CSF 120 min after im injection. Please provide justification for the chosen dose of DCZ.

Lastly, there are a few minor points that should be addressed. Please check all references to figure panels within the main text, in particular those regarding figure 2. The statistical test used in Figure 3 should be revisited. A two-way repeated measures ANOVA would have been a better statistical test for the data represented in Figure 3a in which time and DCZ-treatment are variables. In figure 3, please check the n's as some do not match what is displayed in the figure or figure legend.

Reviewer #3 (Remarks to the Author):

In this manuscript, Miyakawa and colleagues examined the efficacy of using DREADDs to control seizures in a primate model. They used an inhibitory DREADD with DCZ to attempt to alleviate the effects of a chemically induced seizure in the motor cortex. They found evidence that chemogenetic inhibition can reduce the motor and neural symptoms of their chemically induced seizures. The authors conclude from this study that chemogenetic modulation may be a promising treatment for drug-resistant epilepsy, with well laid out caveats about questions that still remain.

Overall, the manuscript presents an exciting and reasonable argument for exploring the use of DREADDs as a novel treatment for drug-resistant epilepsy. Additionally, their data point to the potential effectiveness of DREADDs for a number of possible intractable neurological issues. Despite these positives, there are a number of concerns I have based on some of the experimental and statistical methods employed, that I feel need further addressing. In particular, there seems to a lack of detail in many of their results that diminish my enthusiasm for the current manuscripts, as it appears to be only partially analyzed. Additionally, while this reviewer agrees that NHP studies are critical for any treatment discovery, the authors have not fully explained what their study does to improve upon previous rodent results beyond moving them to NHP. Listed below are some of the concerns and questions I have for the current version of the manuscript.

Major Points:

There is a lack of quantification/statistical comparisons within the results section. For instance, at lines 125 the authors state that there was a significant impairment but don't report any quantification of the behavior or its statistical difference. Without a more full quantification of the effects, it is difficult to interpret the translational potential of the proposed treatment.

Building off this last point, the authors state that the effects of DCZ can be seen within a few minutes but don't quantify what that would actually equate too. They eventually break the data into 5min blocks, however it would be of great interest to see a significantly more fine-grain temporal analysis, as this could be critical to understand the efficacy of DREADDs and DCZ for treating epilepsy as patients require a fast acting effect.

Similarly, they show the spread of and contraction of the electrophysiological effects of the bicuculline in Figures 4 and S5, however there is a lack of details on the temporal effects as they only present and discuss acute snapshots of the responses. For instance, in Figure S5 it would be interesting to understand how the contralateral field spreads and retracts in time. The current data only show a fully spread activation.

The authors report a significant effect of DCZ by running a paired samples t-test between the pre and post injection data. However, this comparison is not the correct statistical test for the data. They should be running a something akin to a GLM or ANOVA with levels for pre and post inject and different drugs (DCZ versus Vehicle). If the full model is significant then a post-hoc comparison of only the DCZ data would show that the DCZ was the cause of the effect seen in the full model. As run the statistics are hard to interpret as they ignore a considerable part of the data.

Minor Points:

Line 115: The authors should specify the AAV used even with it being specified in the methods section. The exact viral vector will be critical for translation to humans.

Reviewer #4 (Remarks to the Author):

This manuscript by Miyakawa and colleagues uses a chemogenetic approach to suppress seizures in a non human primate model. This is significant to the field, in so far as this is the first chemogenetic approach to epilepsy applied in monkeys. The findings generally replicate those previously published in rodent models (I.e., suppression of cortical seizure activity - and subsequent generalized seizures) by chemogenetic inhibition of the seizure focus. (see Katzel et al., 2014 Nat Comm).

The work generally supports the conclusions, and the conclusions are not particularly remarkable in so far as seizures are being suppressed by inhibiting the same site (focus) that is being activated by bicuculline. That said, it is a clear advance to the field to extend this work into primates.

It would be helpful to see more example EEG traces for Figure 2. The spectrogram suggests that there is a change in power during deschlorochlozapine (DCZ) treatment, but activity is still clearly elevated. The small 10 sec traces are not sufficient to fully evaluate these data. It is unclear to me if there is ictal activity continuing during the period of DCZ treatment that is behaviorally silent. I would like to see some continuous EEG traces through the baseline and treatment and post-treatment periods displayed.

I have a fundamental disagreement with the terminology used (eg., Lines 161, 164, 173, 176). The authors refer to effects as anti-epileptic, the seizures as epileptic, the behavior as epileptic. This is simply not the case. This not a model of epilepsy, this is a model of seizures. Epilepsy is a chronic condition, characterized by recurrent seizures. The effects are anti-seizure (or perhaps anticonvulsant).

Line 195: There is insufficient detail to interpret this finding. The number of seizures in which they identified focal-to-bilateral seizure activity, recruitment of a mirror focus, and the fraction of those seizures where the DCZ effect occurred as described is not described.

They describe a dose range of 0.1 to 0.2 mg/kg but it is unclear what data are drawn from which dose, if there were differences between the doses, or if a higher or lower dose would have been effective.

They use a wide range of dose and volume for bicuculline (0.2 to 8 ug), and this was titrated until behavioral seizures were observed. I would like to see a table listing each trial and the outcomes.

There is no justification for performing the experiments under xylazine sedation. The degree to which the effects would hold true in a non0-sedated animal remain a question and need to be discussed.

I would like to see PET data presented from both animals, as the methods say it was acquired from both animals.

Finally, I also have some issues with the data analysis. Treating individual sessions as the unit of analysis fundamentally violates the assumptions of t-tests (paired or otherwise) as the individual sessions do not differ in the site of DREADD injection, nor the location of bicuculline injection. I appreciate the fact that these are non-human primates, that they are a rare and valuable resource, and that these are technically challenging experiments, but also not consistent with best

practices in reporting N-of-1 trial data. N-of-1 data tend to exhibit serial correlation effects and this isn't accounted for (and in fact violates the assumption of independence) of t-tests.

Reviewer Comments:

Reviewer #1 (Remarks to the Author)

This manuscript by Miyakawa et al reports proof of concept experiments showing that DREADDs can be used to attenuate focal epilepsy in nonhuman primates. This is an excellent and very well written manuscript that describes well-designed studies using very sophisticated and state of the art methods and procedures. I only have minor comments.

R: We thank the reviewer for this positive assessment of our manuscript.

The effect of DCZ and hM4Di activation is affecting widespread seizure activity is impressive. What do the authors think is driving this effect? Perhaps they can comment on this a bit more in the discussion.

R: Thank you for the positive comment and for bringing up this point. The most plausible explanation for the chemogenetic effects on widespread seizures is that by suppressing the seizure source at the MI focus, the surrounding suppression network (which are not directly affected by bicuculline) can then begin to function normally, leading to the retraction of widespread seizures in our model. We think that the efficacy of chemogenetic focal silencing for widespread seizure activity needs to be further tested using chronic epilepsy models, in which the mechanisms of seizure propagation are more complex. We have included following discussion of this issue in the revised manuscript.

“These spatially wide-spread seizures, and even the bilaterally spread seizures, were largely attenuated by activation of inhibitory DREADDs around the original epileptic focus, probably because the surrounding inhibitory network (not directly affected by bicuculline) began to function normally.” (Lines 325-329)

“However, with this model we cannot persuade its long-term efficacy in reducing chronic recurring wide-spread seizures of epilepsies. For this, the DREADD system needs to be tested in chronic models of epilepsy, such as a kindling model³⁵, in which the mechanisms of seizure propagation are more complex” (Lines 339-342)

The short-lived efficacy (short relative to its pharmacokinetics) of DCZ in attenuating seizure activity is also interesting and I wonder if the authors would consider also elaborating on this in the

discussion as well.

R: Thank you for pointing out this issue. Together with a similar comment from reviewer #3, we took this point seriously and performed an additional analysis to assess the duration and impact of the anti-seizure effects after the first and second doses of DCZ. The results are provided in a revised figure (Fig. 3) and an additional supplementary figure (Supplementary Fig. 5), which indicate that the effects of the first dose lasted at least 20 minutes in both monkeys. For monkey #1, the effects generally lasted until the end of the daily experiments (about 60 minutes). For monkey #2, the amplitude of the seizures gradually returned, and the effects became less clear 40 minutes after DCZ treatment. The second DCZ dose did not further attenuate seizures in either monkey. On the other hand, attenuation on clonic seizures consistently lasted for the full 60 minutes after DCZ treatment in both monkeys, although this effect occasionally weakened. Therefore, we think it is reasonable to assume that the shorter-lasting seizure reduction in monkey #2 was somehow due to a difference between the two monkeys. Indeed, when we inspected hM4Di expression in the monkeys, we found it was much lower in the superficial layer of MI in monkey #2 than in monkey #1, as evidenced by GFP immunohistochemistry. Because of this, the return of seizures in monkey #2 during the later sessions might have resulted from this reduced hM4Di expression combined with bicuculline diffusion into the superficial layers over time. This interpretation is consistent with the findings regarding clonic seizures; the deep layer (the output layer) of MI in both monkeys sufficiently expressed hM4Di and thus its output was continuously suppressed by DCZ, which led to stable attenuation of convulsive behavior for 60 minutes. We included the discussion on this issue as follows:

“While the anti-convulsant effects of DCZ were roughly consistent throughout the 1-h session, the anti-seizure effect in the MI of monkey #2 only lasted about 20 min. This is not likely related to a rapid decrease in DCZ because the second dose of DCZ had no additional impact on either seizures or convulsions. A more plausible reason for the short-lived effect is that monkey #1 and monkey #2 differed in some critical way. Indeed, GFP immunohistochemistry in monkey #2 indicated that hM4Di expression in the superficial layer of MI was much lower than in the deep layer. In contrast, hM4Di expression in the superficial and deep layers of monkey #1 was equally dense (cf. Fig. 5 and Supplementary Fig. 1). The low expression in the superficial layers of monkey #2 might have led to the return of cortical seizures in the later sessions when bicuculline could have diffused to the superficial layers. This view is consistent with results for

clonic seizures; because the deep layers of MI in both monkeys expressed enough hM4Di, its output was suppressed by DCZ administration, which led to continuously attenuated convulsive behavior for 1 h.” (Lines 302-315)

Line 87: please consider changing to “which can have”

R: Thank you. We have changed the text as suggested.

Figure 1: “epilepsy” is misspelled

R: Thank you, we’ve corrected the spelling.

Line 240: consider changing to “viral vectors”

R: Thank you. We have changed the text as suggested.

Line 302: please delete the second “the”

R: Deleted. Thank you.

Reviewer #2 (Remarks to the Author)

The manuscript by Miyakawa et al. titled “Chemogenetic attenuation of cortical seizures in primates” is a proof of principal study. The main goal of this study was to demonstrate the feasibility of chemogenetic technology as an alternative treatment for frontal lobe epilepsy. Using 1-2 cynomolgus macaques, this group used a pharmacologically-induced model of epilepsy in combination with inhibitory-DREADDs, positron emission tomography, and electrocorticogram. Monkeys had hM4Di introduced into the left primary motor cortex which later served as the same location for the seizure focus by the administration of a pharmacological agent, bicuculline. The main conclusion of the study shows that hM4Di is an effective way to dampen seizure severity only if expressed within the seizure focus. The premise of this manuscript has potential for high clinical impact, but the study has some issues that should be addressed.

R: We thank the reviewer for this encouraging assessment.

Although it is understandable the difficulty and limited resources involved in NHP research, the

feasibility of this study would be greatly strengthened by control studies in which seizure within the M1 region is elicited, behavioral, and EEG studies are performed in NHPs that do not express hM4Di. This will be important for immunohistological analysis of cell death.¹⁾ In addition to a comparison of outcome measures between hM4Di-activated and hM4Di non-expressing NHPs, the study would be strengthened by a comparison of the two hM4Di-expressing NHPs. There were differences between the 2 NHPs in their AAV viral titer used, the number of seizures elicited, potential damage caused by surgery from placement of ECoG channels bilaterally versus unilaterally. Additionally, it is unclear if the two hM4Di-NHPs were used for each experiment (i.e., behavior, PET scans to examine location of DREADD expression, ECoG measurements during seizure, seizure, immunohistochemistry). For example, page 6 line 125 (Sup. Fig. 1) shows that chemogenetic function of hM4i expressed in the M1 by testing grasping was only performed in monkey #2. The use of a table would be necessary to understand the use of each animal and to determine if the information presented is valid across animals.²⁾ This would also allow for the much-needed clarification of method details such as how many times a seizure was elicited in each NHP, the order of treatment (ie DCZ or vehicle).

R: Thank you for these comments and suggestions. We'll address the points in order. 1) We agree with the reviewer that control studies are important for determining whether the effects of DCZ depend on the DREADD, and whether overexpression of the DREADD leads to tissue damage or cell death. As we have described in the original manuscript, a control study was performed using monkey #2. In two sessions, bicuculline was injected into the contralateral M1 region that did not express hM4Di. Seizures were not attenuated by DCZ treatment in either session, confirming that the effect of DCZ was specific to the seizures induced in cortical regions that expressed hM4Di (Supplementary Fig. 6). In these cases, we ended up administering antiepileptic drugs (Diazepam) to control the seizures. However, there remains concern that long-lasting seizures could damage other brain regions as well as the non-DREADD MI.

To compare the effects of seizure induction (bicuculline injection) with and without hM4Di expression in brain tissue, we performed additional immunohistological analysis of immune responses and cell loss in monkey #2. The results are shown in Supplementary Fig. 1. In a NeuN-stained section, moderate cell loss was observed in the deeper layers of the non-DREADD side of MI. Immune responses were found along the needle tracks, as indicated by strong positive signals for astrocyte marker (GFAP) in both DREADD and non-DREADD MI. We did not observe any cell loss or immune response in regions overlapping with the hM4Di expression. Thus, cell loss and

immunoreactivity appears to have been due to bicuculline injection and seizures, rather than being induced by overexpression and/or activation of hM4Di. In the revised manuscript, we emphasize the results of this control experiment in the Results as follows:

“We obtained consistent immunohistological observations from monkey #2, who had DREADD expression in the left hemisphere and ECoG electrode implantation and bicuculline injections in both hemispheres (Table 1). Moderate neuronal loss was observed in the deep layers of the right MI (Supplementary Fig. 1d) and was not colocalized with the GFP positive regions (Supplementary Fig. 1c, 1d, 1g, 1h). Prominent immune responses were found along the needle tracks as visualized by a strong Glial Fibrillary Acidic Protein (GFAP, astrocyte marker) signal (Supplementary Fig. 1e, 1i).” (Lines 241-248)

Certainly, as the reviewer pointed out, a control study using non-DREADD monkey(s) might strengthen our observations. However, the need for this control should be considered from an ethical standpoint, especially since the anticonvulsant effect of DCZ would not be expected in these cases. We believe that our control experiments described above allowed us to confirm the DREADD-selective effects of DCZ and the safety of DREADD with the advantage of a within-subject comparison that can eliminate the effects of uncontrollable individual differences.

2) As the reviewer suggested, we have added a summary of the experimental conditions and tests for each monkey in Table 1. This will provide readers with a clear overview of how the experimental conditions differed between the monkeys (titers of AAV vector, number of bicuculline injections, and number of DCZ/vehicle treatments), and which experiments were performed for each monkey. For example, we now refer to this table in the main text regarding the grasping task test of chemogenetic function in monkey #2:

“We verified the chemogenetic function of hM4Di expressed in the MI by testing the grasping ability of a monkey (monkey #2; Table 1).” (Lines 125-126)

We have also added a summary of bicuculline doses and timing of vehicle and DCZ injections in Supplementary Table 1. A detailed description of treatment with reference to the table is now in the Methods.

“The first injection of DCZ was delivered 15–20 min after bicuculline (1 case) or vehicle (5 cases). The second injection of DCZ was delivered 18–30 min after the first one (5 cases). The timing of vehicle and DCZ administration for each session is summarized in Supplementary Table 1.” (Lines 509-512)

Secondly, a discussion or additional data regarding the quick onset and offset of DCZ action is needed. It seems a bit fast for the onset of DCZ action within the brain, as observed by a decrease in seizure amplitude that occurred within a 1 minute after an intramuscular injection. Additionally, the effect of DCZ ended after only 25 minutes when the same group has shown high binding within the putamen 90min after injection and similar concentrations of DCZ within the CSF 120 min after im injection. Please provide justification for the chosen dose of DCZ.

R: Rapid onset of chemogenetic effects on neuronal activity after DCZ administration has already been demonstrated using an hM3Dq-DREADD-expressing monkey (Nagai et al., Nat Neurosci. 2020). Our new analysis with finer temporal resolution demonstrated the rapid decrease in seizure amplitude (within 1 minute after intramuscular DCZ injection, Fig. 3a), which was in accordance with this previous finding and thus appears to reflect the fast pharmacokinetics. We now mention this and cite the relevant papers:

“Our data indicate that systemic administration of DCZ is effective for suppressing cortical and clonic seizures within 1 and 3 min, respectively. The rapid chemogenetic action of DCZ is consistent with previous reports and will be extremely useful in prospective therapeutic applications. Here, we used the same DCZ dose (0.1 mg kg⁻¹) as in previous reports^{12,30,31} ...” (Lines 288-293)

A new analysis (Supplementary Fig. 5a) indicates that the significant anti-seizure effect lasted for 60 minutes in monkey #1 but only for about 20 minutes in monkey #2. Added to that, a comparison of first and second DCZ doses showed that the second dose had no significant effect on seizure attenuation (Supplementary Fig. 5b), indicating that the chemogenetic anti-seizure effect had reached its maximum level. However, unlike this difference for cortical seizures, the anticonvulsant effect of DCZ on clonic seizures generally lasted for 60 minutes in both monkeys, although it occasionally weakened (Supplementary Fig. 5c). Thus, the shorter duration of the anti-seizure effect

in monkey #2 may be due to some difference between the two monkeys. In fact, we found a major difference in hM4Di expression, which was much lower in the superficial layer of MI in monkey #2 than in monkey #1, as evidenced by GFP immunohistochemistry. Because of this, seizures in monkey #2 might have returned in the later session if bicuculline diffused into the superficial layers. This interpretation is consistent with the findings regarding clonic seizures; the deep layer (the output layer) of MI in both monkeys sufficiently expressed hM4Di and thus its output was continuously suppressed by DCZ, which led to stable attenuation of convulsive behavior for 60 minutes. We included the discussion on this issue as follows:

“While the anti-convulsant effects of DCZ were roughly consistent throughout the 1-h session, the anti-seizure effect in the MI of monkey #2 only lasted about 20 min. This is not likely related to a rapid decrease in DCZ because the second dose of DCZ had no additional impact on either seizures or convulsions. A more plausible reason for the short-lived effect is that monkey #1 and monkey #2 differed in some critical way. Indeed, GFP immunohistochemistry in monkey #2 indicated that hM4Di expression in the superficial layer of MI was much lower than in the deep layer. In contrast, hM4Di expression in the superficial and deep layers of monkey #1 was equally dense (cf. Fig. 5 and Supplementary Fig. 1). The low expression in the superficial layers of monkey #2 might have led to the return of cortical seizures in the later sessions when bicuculline could have diffused to the superficial layers. This view is consistent with results for clonic seizures; because the deep layers of MI in both monkeys expressed enough hM4Di, its output was suppressed by DCZ administration, which led to continuously attenuated convulsive behavior for 1 h.” (Lines 302-315)

We also provide justification for the chosen dose of DCZ in the Methods.

“Note that this dose of DCZ yields 80%–90% hM4Di occupancy and affects behavioral performance via hM4Di activation in monkeys¹².” (Lines 505-507)

Lastly, there are a few minor points that should be addressed. Please check all references to figure panels within the main text, in particular those regarding figure 2. The statistical test used in Figure 3 should be revisited. A two-way repeated measures ANOVA would have been a better statistical test for the data represented in Figure 3a in which time and DCZ-treatment are variables. In figure 3,

please check the n's as some do not match what is displayed in the figure or figure legend.

R: We have checked that all references to figure panels in the main text are now correct. We thank the reviewer for pointing this out. Regarding the statistics used in Figure 3, together with related comments from reviewers #3 and #4, we revised the statistics and visualization for all panels so that the onset and the strength of treatment effects were analyzed separately. Figure 3a represents the effects of treatment on cortical seizure in time, and significant effects were defined as a deviation from the 95th percentile of the values obtained during baseline. Fig. 3b provides a direct comparison of the effect between DCZ and vehicle by using a repeated-measures two-way ANOVA. We modified the Results accordingly as follows:

“We quantified the onset and strength of the anti-seizure effects for DCZ (6 sessions in total from the two monkeys) and compared them with those for the vehicle control (4 sessions). At 1 min after DCZ administration, the mean seizure amplitude decreased and remained below the 95th percentile of what it was during the baseline period. In contrast, it did not change after vehicle administration (Fig. 3a). The effect of treatment on seizure amplitudes differed significantly between DCZ and vehicle (two-way repeated-measures ANOVA: treatment × time interaction: $F_{(1, 11)} = 42.98$, $p = 1.7 \times 10^{-4}$) such that DCZ clearly triggered a drop in amplitude (Fig. 3b).” (Lines 165-173)

Because the number of sessions for DCZ and vehicle administration is now summarized in Supplementary Table 1, we referred to the table in the legend of Fig. 3 and confirmed that they are consistent.

Reviewer #3 (Remarks to the Author)

In this manuscript, Miyakawa and colleagues examined the efficacy of using DREADDs to control seizures in a primate model. They used an inhibitory DREADD with DCZ to attempt to alleviate the effects of a chemically induced seizure in the motor cortex. They found evidence that chemogenetic inhibition can reduce the motor and neural symptoms of their chemically induced seizures. The authors conclude from this study that chemogenetic modulation may be a promising treatment for drug-resistant epilepsy, with well laid out caveats about questions that still remain.

Overall, the manuscript presents an exciting and reasonable argument for exploring the use of DREADDs as a novel treatment for drug-resistant epilepsy. Additionally, their data point to the potential effectiveness of DREADDs for a number of possible intractable neurological issues. Despite these positives, there are a number of concerns I have based on some of the experimental and statistical methods employed, that I feel need further addressing. In particular, there seems to be a lack of detail in many of their results that diminish my enthusiasm for the current manuscripts, as it appears to be only partially analyzed. Additionally, while this reviewer agrees that NHP studies are critical for any treatment discovery, the authors have not fully explained what their study does to improve upon previous rodent results beyond moving them to NHP. Listed below are some of the concerns and questions I have for the current version of the manuscript.

R: We thank the reviewer for this greatly encouraging assessment and the constructive comments that followed.

Major Points:

There is a lack of quantification/statistical comparisons within the results section. For instance, at lines 125 the authors state that there was a significant impairment but don't report any quantification of the behavior or its statistical difference. Without a more full quantification of the effects, it is difficult to interpret the translational potential of the proposed treatment.

R: As the reviewer pointed out, the statistical results originally shown in Supplementary Fig. 1 (now Supplementary Fig. 2) is now reported in the main text as follows:

“Systemic administration of the DREADD actuator DCZ (0.1 mg kg⁻¹, intramuscular injection) significantly and reproducibly impaired food retrieval using the contralateral (right) hand—but not the ipsilateral (left) hand—compared with vehicle control treatment (paired t-test; contralateral: $p = 0.0097$; ipsilateral: $p = 0.74$; Supplementary Fig. 2).”
(Lines 126-130)

Building off this last point, the authors state that the effects of DCZ can be seen within a few minutes but don't quantify what that would actually equate to. They eventually break the data into 5min blocks, however it would be of great interest to see a significantly more fine-grain temporal analysis, as this could be critical to understand the efficacy of DREADDs and DCZ for treating

epilepsy as patients require a fast acting effect.

R: Thank you for this suggestion. To quantify the latency of the DCZ effect on cortical seizures and bodily convulsions with a finer time scale, we revised the temporal analysis that was originally shown in Figure 3a and 3c. We considered a change to be significant if the measured value deviated from the 95th percentile of what was observed at baseline, as shown in Figure 3a and 3c. In this analysis, the temporal resolution was set to 1 min. The results represent a fast-acting effect of DCZ, with cortical seizures and convulsions significantly reduced at least 1 and 3 min after DCZ administration, respectively. Results and Methods were revised accordingly as follows:

“We quantified the onset and strength of the anti-seizure effects for DCZ (6 sessions in total from the two monkeys) and compared them with those for the vehicle control (4 sessions). At 1 min after DCZ administration, the mean seizure amplitude decreased and remained below the 95th percentile of what it was during the baseline period. In contrast, it did not change after vehicle administration (Fig. 3a). The effect of treatment on seizure amplitudes differed significantly between DCZ and vehicle (repeated-measures two-way ANOVA; treatment \times time interaction: $F_{(1, 11)} = 42.98$, $p = 1.7 \times 10^{-4}$) such that DCZ clearly triggered a drop in amplitude (Fig. 3b). Furthermore, administering DCZ, but not the vehicle, rapidly attenuated clonic seizures (convulsions in the body, head, and arms), which remained below the baseline range within 3 min (Fig. 3c).” (Lines 165-175)

“To quantify the effects of DCZ and vehicle treatment on cortical seizure, the amplitude was first calculated as the RMS of the waveform in each 1 min bin before (-15 to 0 min) and after treatment (0 to 60 min and 0 to 15 min for DCZ and vehicle, respectively). Then the values were normalized by the average value during each session’s pre-administration period. We considered a change to be significant if the measured value deviated from the 95th percentile of the values during pre-administration period. For display purposes, the results were then scaled by the maximum absolute values for the pre- and post-administration periods (-15 to 12 min). The average values during the pre-administration (-6 to -2 min) and post-administration (2 to 6 min) periods were compared between treatment conditions using a two-way (treatment \times time) repeated-measures ANOVA. For clonic seizures, the frequency of convulsions was calculated for

each 1 min bin and smoothed with 5-min windows shifted by 1 min. The other procedures were the same as those for cortical seizures.” (Lines 599-612)

Similarly, they show the spread of and contraction of the electrophysiological effects of the bicuculline in Figures 4 and S5, however there is a lack of details on the temporal effects as they only present and discuss acute snapshots of the responses. For instance, in Figure S5 it would be interesting to understand how the contralateral field spreads and retracts in time. The current data only show a fully spread activation.

R: Thank you for this suggestion. We have revised Supplementary Fig. 5 (now Supplementary Fig. 8) to show how the seizure spread and retracted in the contralateral cortical field over time. We additionally provide the details for the temporal and spatial dynamics of seizure spread and DCZ effects by referring to this figure as follows:

“In monkey #2, who had ECoGs implanted in both hemispheres, seizures in similar conditions spread from the induced area to the opposite hemisphere (Supplementary Fig. 8b-e, -ii to -v). Concurrently, clonic seizures also spread to body parts governed by the non-DREADD hemisphere (e.g., left hand; Fig. 2b; Supplementary Fig. 8g). Remarkably, the anti-seizure effect of DCZ was not limited to the DREADD-expressing region but extended to the entire recording region after the seizure had spread (Fig. 4ab-v to -vii). Specifically, DREADD/DCZ-induced inactivation in the seizure focus was sufficient to reverse seizures even after focal-to-bilateral tonic clonic seizure has occurred (e.g., Supplementary Fig. 8c-e, -vi to -iix; all four cases in monkey #2, see Supplementary Table 2).” (Lines 216-226)

The authors report a significant effect of DCZ by running a paired samples t-test between the pre and post injection data. However, this comparison is not the correct statistical test for the data. They should be running a something akin to a GLM or ANOVA with levels for pre and post inject and different drugs (DCZ versus Vehicle). If the full model is significant then a post-hoc comparison of only the DCZ data would show that the DCZ was the cause of the effect seen in the full model. As run the statistics are hard to interpret as they ignore a considerable part of the data.

R: Following the reviewer's advice, we directly compared the effects of different treatments (i.e., DCZ vs. vehicle) on cortical seizures and convulsive behavior using two-way repeated measures

ANOVA (treatment x time) as shown in Fig. 3b and 3d. We revised the Results and Methods accordingly:

“The effect of treatment on seizure amplitudes differed significantly between DCZ and vehicle (two-way repeated-measures ANOVA; treatment × time interaction: $F_{(1, 1)} = 42.98$, $p = 1.7 \times 10^{-4}$) such that DCZ clearly triggered a drop in amplitude (Fig. 3b). Furthermore, administering DCZ, but not the vehicle, rapidly attenuated clonic seizures (convulsions in the body, head, and arms), which remained below the baseline range within 3 min (Fig. 3c). In addition to declining in the right hand, which is controlled by the target left MI cortex (58% decline; Supplementary Fig. 4b left), clonic seizures also declined in other parts of the body (99% decline, Supplementary Fig. 4c, left). As with seizure activity in the brain, only DCZ significantly affected clonic seizures (two-way repeated-measures ANOVA; treatment × time interaction: $F_{(1, 10)} = 30.96$, $p = 8.4 \times 10^{-4}$), again triggering a clear drop in the amplitude (Fig. 3d).” (Lines 170-180)

Minor Points:

Line 115: The authors should specify the AAV used even with it being specified in the methods section. The exact viral vector will be critical for translation to humans.

R: As suggested, we have now specified the AAV used.

“To focally express hM4Di, we used a mosaic adeno-associated virus vector (AAV2.1-hSyn-FLAG-hM4Di-IRES-AcGFP) for neuron-specific expression of hM4Di and a fluorescent marker.” (Lines 114-117)

We also provide additional information about the production of the AAV2.1 vector in the Methods.

“For production of the AAV2.1 vector, the pAAV-RC1 plasmid-coding AAV1 capsid protein and the pAAV-RC2 plasmid-coding AAV2 capsid protein were transfected in a 1:9 ratio.” (Lines 383-385)

Reviewer #4 (Remarks to the Author):

This manuscript by Miyakawa and colleagues uses a chemogenetic approach to suppress seizures in a non human primate model. This is significant to the field, in so far as this is the first chemogenetic approach to epilepsy applied in monkeys. The findings generally replicate those previously published in rodent models (I.e., suppression of cortical seizure activity - and subsequent generalized seizures) by chemogenetic inhibition of the seizure focus. (see Katzel et al., 2014 Nat Comm).

The work generally supports the conclusions, and the conclusions are not particularly remarkable in so far as seizures are being suppressed by inhibiting the same site (focus) that is being activated by bicuculline. That said, it is a clear advance to the field to extend this work into primates.

R: We would like to thank the reviewer for the careful review and constructive comments regarding our manuscript.

It would be helpful to see more example EEG traces for Figure 2. The spectrogram suggests that there is a change in power during deschlorochlozapine (DCZ) treatment, but activity is still clearly elevated. The small 10 sec traces are not sufficient to fully evaluate these data. It is unclear to me if there is ictal activity continuing during the period of DCZ treatment that is behaviorally silent. I would like to see some continuous EEG traces through the baseline and treatment and post-treatment periods displayed.

R: As the reviewer requested, we have now provided 1-min ECoG traces in an additional supplementary figure (Supplementary Fig. 3), allowing for evaluation of inter-ictal activity during the DCZ treatment period (panel v). We also provide several continuous ECoG traces for sessions 1 and 2 in monkey #1, so that the ECoG data shown in Fig. 2 can be evaluated through the pre- and post-DCZ administration periods. We refer to the figures in the main text as follows:

“Immediately after bicuculline infusion, spectral power lower than 20 Hz initiated an upward trend (Fig. 2b), reflecting epileptic “spikes”, “spike-wave complexes”, and “multi-spike-wave complexes” (or “poly-spikes”) (Fig. 2a-ii, iii; Supplementary Fig. 3a for a longer time period), which are stereotypical of preliminary and mild epileptic seizures²¹.” (Lines 142-146)

“DCZ injection resulted in a rapid decrease in seizure amplitude, disappearance of multi-spike complexes, and concomitant decrease in clonic seizures (Fig. 2a-v, 2b; Supplementary Fig. 3a-v; 3b). Although seizure discharges and clonic seizures occasionally recurred after DCZ injection (e.g., ~25 min after the first DCZ injection, Fig. 2b), seizure amplitudes were lower and clonic seizures were fewer compared with what was observed following sham treatments (i.e., vehicle) (Fig. 2c). Moreover, recurring cortical seizures and bodily or bilateral clonic seizures were clearly attenuated after administering DCZ (Fig. 2b, 2c, yellow arrowhead; Supplementary Fig. 3c). (Lines 157-165)

I have a fundamental disagreement with the terminology used (eg., Lines 161, 164, 173, 176). The authors refer to effects as anti-epileptic, the seizures as epileptic, the behavior as epileptic. This is simply not the case. This not a model of epilepsy, this is a model of seizures. Epilepsy is a chronic condition, characterized by recurrent seizures. The effects are anti-seizure (or perhaps anticonvulsant).

R: we understand the issue. According to the reviewer’s advice, we have revised the terminology throughout the manuscript. For example, we use ‘anti-convulsant’ or ‘anti-seizure’ instead of anti-epilepsy, and ‘bodily convulsions’ instead of epileptic body movements.

Line 195: There is insufficient detail to interpret this finding. The number of seizures in which they identified focal-to-bilateral seizure activity, recruitment of a mirror focus, and the fraction of those seizures where the DCZ effect occurred as described is not described.

R: We have now summarized the results for the fraction of focal-to-bilateral seizures in Supplementary Table 2 by describing the types of seizure waveforms induced by bicuculline and reduced by DCZ injections. We have revised the insufficient description of the results that the reviewer mentions by referring to a typical example of ECoG activity maps representing how the seizures spreads and retracts in the contralateral field over time (Supplementary Fig. 8). We also refer to the new Supplementary Table 2, as follows:

“Remarkably, the anti-seizure effect of DCZ was not limited to the DREADD-expressing region but extended to the entire recording region after the seizure had spread (Fig. 4a-v to -vii). Specifically, DREADD/DCZ-induced inactivation in the seizure focus was

sufficient to reverse seizures even after focal-to-bilateral tonic clonic seizure has occurred (e.g., Supplementary Fig. 8b-e, -vi to -iix; all four cases in monkey #2, see Supplementary Table 2).” (Lines 220-226)

They describe a dose range of 0.1 to 0.2 mg/kg but it is unclear what data are drawn from which dose, if there were differences between the doses, or if a higher or lower dose would have been effective.

R: We apologize for the lack of detail. We injected a fixed dose of DCZ (0.1 mg kg⁻¹) once or twice during each treatment session separately with 18–30 min intervals between injections. We quantified the effect of the first and second DCZ doses in Fig. 3 and Supplementary Fig. 5, respectively. We summarize the sequence and timing of DCZ and vehicle treatments in Supplementary Table 1 and clarify them in the Results and Methods.

“After seizures became behaviorally visible, we administered DCZ alone, or the vehicle followed by DCZ, intramuscularly (Supplementary Table 1).” (Lines 156-157)

“The first injection of DCZ was delivered 15–20 min after bicuculline (1 case) or vehicle (5 cases). The second injection of DCZ was delivered 18–30 min after the first one (5 cases). The timing of vehicle and DCZ administration for each session is summarized in Supplementary Table 1.” (Lines 509-512)

They use a wide range of dose and volume for bicuculline (0.2 to 8 ug), and this was titrated until behavioral seizures were observed. I would like to see a table listing each trial and the outcomes.

R: As suggested, we have now included Supplementary Table 1, which summarize the dose and volume of bicuculline used in each session.

There is no justification for performing the experiments under xylazine sedation. The degree to which the effects would hold true in a non-sedated animal remain a question and need to be discussed.

R: In order to monitor and evaluate convulsive body movements and to distinguish them from voluntary movements in awake animals, it is necessary to conduct experiments under sedated/immobilized conditions. We provide this justification in the Methods as follows.

“To monitor and evaluate convulsive body movements and to distinguish them from voluntary movements, monkeys were sedated with intramuscular injection of xylazine (0.8–1.2 mg kg⁻¹ initial dose, and 0.4–0.8 mg kg⁻¹ h⁻¹ for maintenance).” (Lines 476-479)

We also added the following limitation of this study:

“Second, the current study was performed under xylazine sedation because bodily convulsions had to be clearly distinguished from voluntary movements. Future studies are needed to directly determine the extent to which convulsions are suppressed in non-sedated animals. (Lines 343-346)

I would like to see PET data presented from both animals, as the methods say it was acquired from both animals.

R: In addition to the PET data for monkey #1 shown in Fig. 1, we now provide Supplementary Fig. 1a, which shows the PET data visualizing hM4Di expression in MI that corresponded to the region of high GFP-positive signal that was found *in vitro* for monkey #2. We refer to the figure in the main text as follows:

“We observed a prominent PET signal, indicating successful hM4Di expression at the target region (Fig. 1c, d; Supplementary Fig. 1a), which was confirmed by post-mortem immunohistochemical staining with an antibody for the co-expressed AcGFP (Fig. 1e; Supplementary Fig. 1b).” (Lines 121-124)

Finally, I also have some issues with the data analysis. Treating individual sessions as the unit of analysis fundamentally violates the assumptions of t-tests (paired or otherwise) as the individual sessions do not differ in the site of DREADD injection, nor the location of bicuculline injection. I appreciate the fact that these are non-human primates, that they are a rare and valuable resource, and that these are technically challenging experiments, but also not consistent with best practices in reporting N-of-1 trial data. N-of-1 data tend to exhibit serial correlation effects and this isn't accounted for (and in fact violates the assumption of independence) of t-tests.

R: We believe that it is generally accepted in neuroscience research using non-human primates to carry out measurements repeatedly from the same individuals and consider them as independent measurements, thus satisfying the assumptions of ANOVA. A model with mixed effects for both individuals and measurements will increase the number of independent variables relative to the

sample size ($N = 4\sim 5$), making it difficult to accurately test the null hypothesis. Thus, as suggested by reviewer #3, we have now analyzed the data using a repeated-measures design, taking into account the correspondence between vehicle and DCZ administration.

Peer review comments,

Second round review:

Reviewer #1 (Remarks to the Author):

The authors have done a great job to address my comments. Thank you.

Reviewer #2 (Remarks to the Author):

This is a revised manuscript titled "Chemogenetic attenuation of cortical seizures in primates" by Miyakawa and colleagues. The study used a combination of techniques in this study to demonstrate the feasibility of using inhibitory chemogenetics to alleviate seizures that included MRI, CT, PET, and ecog. In 2 cynomolgus monkeys, the stereotaxically injected hM4Di unilaterally into the primary motor cortex of the left hemisphere. After recovery and allowing time for viral expression, seizures were induced using the GABA-A receptor antagonist, bicuculline. This study has potential for high clinical impact and is suitable for publication in Nature Communications. There are a few very minor points that could be addressed.

On page 6 first paragraph last sentence, mentions "that DCZ alone did not affect cognitive or motor ability". However, cognitive function was not assessed and therefore please delete cognitive. On page 10 last paragraph, it states that "we obtained consistent immunohistological observations from monkey #2." However, with the differences in titer used between the two animals, it is hard to determine if the larger titer is associated with an increase in cell death without statistics or images to compare the two monkeys. Lastly, the fluorescent images in Supplementary figure 1 are very hard to see.

Reviewer #3 (Remarks to the Author):

The authors, on the whole, have done an excellent job of addressing my comments from the previous draft. Overall, I feel that the manuscript is significantly improved and more clear. I do have two remaining comments/concerns that I feel should be addressed:

- 1) Most importantly, since the main difference in the anti-seizure effect has been associated with differences in the anatomy of the two animals (Discussion lines 302-315), the histological results on monkey 2 should be presented in addition to M1's so readers can evaluate these differences for themselves.
- 2) Figure 3b and 3d: for clarity Vehicle and DCZ should be plotted on the same X point for pre and then for post. The offset X points make the timing confusing. The color of two points are enough to differentiate them, in this reviewer's opinion.

Reviewer #4 (Remarks to the Author):

The authors have addressed some of my prior concerns, but others are not satisfactorily addressed. As it stands, this manuscript still reads more like an extended abstract with preliminary findings, rather than a full, well controlled study.

First: The representative EEG traces are not compelling. There is clear ictal activity ongoing during both the control and DCZ periods in both animals.

Second: There is no systematic pattern to the bicuculline doses used. (Suppl. Table 1). How does this influence the results? In some cases quite high doses were used, in others very low doses.

Third: The fact that the experiments were performed under sedation remains concerning. The clinical and translational relevance is significantly reduced by this.

Fourth: I still disagree with the statistical approach. It remains (regardless of protests to the contrary) a violation of the assumptions of independence to analyze these data by T-Test or ANOVA. It doesn't matter if it is "generally accepted" or not - it violates the assumption of independence to treat multiple observations from a single subject as independent observations. They clearly are not. Just because others violate the statistical assumptions doesn't mean it should be allowed to continue. They either need to use a mixed model approach or an alternative approach.

VIEWER COMMENTS

Reviewer #2 (Remarks to the Author):

This is a revised manuscript titled “Chemogenetic attenuation of cortical seizures in primates” by Miyakawa and colleagues. The study used a combination of techniques in this study to demonstrate the feasibility of using inhibitory chemogenetics to alleviate seizures that included MRI, CT, PET, and ecog. In 2 cynomolgus monkeys, the stereotaxically injected hM4Di unilaterally into the primary motor cortex of the left hemisphere. After recovery and allowing time for viral expression, seizures were induced using the GABA-A receptor antagonist, bicuculline. This study has potential for high clinical impact and is suitable for publication in Nature Communications. There are a few very minor points that could be addressed.

R: We thank the reviewer for the positive evaluation. We have addressed each comments bellow.

On page 6 first paragraph last sentence, mentions “that DCZ alone did not affect cognitive or motor ability”. However, cognitive function was not assessed and therefore please delete cognitive.

R: Thank you for pointing this out. We have deleted the word “cognitive” as suggested (Page 6, lines 132)

On page 10 last paragraph, it states that “we obtained consistent immunohistological observations from monkey #2.” However, with the differences in titer used between the two animals, it is hard to determine if the larger titer is associated with an increase in cell death without statistics or images to compare the two monkeys.

R: According to the reviewer’s suggestion, the revised manuscript now includes the PET imaging data and immunohistochemistry photographs (Supplementary Fig. 1), allowing the results from the two monkeys to be directly compared. The comparison indicates that hM4Di expression level was relatively lower in monkey #2 than in monkey #1. We have added a related discussion in the Discussion section.

“Indeed, the PET signal in monkey #2 was lower than that in monkey #1 (Supplementary Fig. 1a), indicating relatively weak hM4Di expression in this monkey. In addition, GFP immunohistochemistry in monkey #2 indicated that hM4Di expression in the superficial layer of MI was lower than in the deep layer.”

Given the small sample size, our study could not quantitatively address the reasons for these observed differences in hM4Di expression levels or expression patterns. However, to answer the reviewer’s specific question, we can at least say that the differences could not be explained by the differences in titers, since we found no clear cell-death or immune response related signal solely related to DREADD expression in either monkey.

Although Supplementary Fig. 1 allows direct comparison of hM4Di expression between the two monkeys, the original sentence highlighted by the reviewer does seem somehow overly simplified. We have therefore revised as follows it to avoid any misconceptions:

“Similarly, immunohistological observations from monkey #2, who had DREADD expression in the left hemisphere, and ECoG electrode implantation and bicuculline injections in both hemispheres (Table 1) showed no clear cell-death or immune response related signal caused by DREADD expression.”

Lastly, the fluorescent images in Supplementary figure 1 are very hard to see.

R: We revised the immunofluorescent images (Supplementary Fig. 9 in the revision) to make them easier to see.

Reviewer #3 (Remarks to the Author):

The authors, on the whole, have done an excellent job of addressing my comments from the previous draft. Overall, I feel that the manuscript is significantly improved and more clear. I do have two remaining comments/concerns that I feel should be addressed:

R: We thank the reviewer for the positive evaluation and constructive comments. We have addressed the new comments/concerns below.

1) Most importantly, since the main difference in the anti-seizure effect has been associated with differences in the anatomy of the two animals (Discussion lines 302-315), the histological results on monkey 2 should be presented in addition to M1's so readers can evaluate these differences for themselves.

R: According to the reviewer's suggestion, and a similar comment from reviewer #2, we now provide additional images in Supplementary Fig. 1 that allow hM4Di expression to be directly compared between the two monkeys.

2) Figure 3b and 3d: for clarity Vehicle and DCZ should be plotted on the same X point for pre and then for post. The offset X points make the timing confusing. The color of two points are enough to differentiate them, in this reviewer's opinion.

R: Thank you for this suggestion. We now plot the data for vehicle and DCZ on the same X points in Figure 3b and 3d, which have been further revised according to comments from reviewer #4.

Reviewer #4 (Remarks to the Author):

The authors have addressed some of my prior concerns, but others are not satisfactorily addressed. As it stands, this manuscript still reads more like an extended abstract with preliminary findings, rather than a full, well controlled study.

R: We appreciate the reviewer's insights and constructive comments regarding on our manuscript and its revisions. We believe that our experimental data, including adequately performed control experiments, provide compelling evidence for chemogenetic attenuation of cortical seizures in monkeys.

First: The representative EEG traces are not compelling. There is clear ictal activity ongoing during both the control and DCZ periods in both animals.

R: Although a few of the EEG traces indeed indicate that some residual ictal activity remained following DCZ treatment (e.g., Supplementary Fig. 3c), most EEG traces demonstrate that ictal activity has disappeared after DCZ (e.g., Fig. 2a; Supplementary Fig. 3ab). Quantitatively, we have shown that DCZ treatment significantly reduced EEG amplitudes, providing clear and compelling evidence for chemogenetic attenuation of seizure amplitude.

Second: There is no systematic pattern to the bicuculline doses used. (Suppl. Table 1). How does this influence the results? In some cases quite high doses were used, in others very low doses.

R: The dose of bicuculline differed from session to session because the same dose did not always induce clonic seizures. We increased it when seizures were absent or weak. A similar strategy has been employed in previous studies with monkeys (e.g., McCairn et al., J. Neurophysiol, 2013; McCairn and Nagai et al., Neuron 2016; Aupy et al., Cerebral Cortex 2020).

To directly address the reviewer's concern with regards to how this might affect the results, we examined whether the dose of bicuculline influenced the outcome following DCZ treatment. We found no significant effects of bicuculline dosage on chemogenetic action in either cortical seizure (linear regression analysis, $F[1,4] = 3.06$, $p = 0.16$) or clonic seizures ($F[1,3] = 5.64$, $p = 0.098$).

We have noted this in the legend of Supplementary Table 1.

Third: The fact that the experiments were performed under sedation remains concerning. The clinical and translational relevance is significantly reduced by this.

R: We understand this concern. All translational research has certain limitations, and we have discussed some limitations of our study, including the issue of sedation. However, given the unproven status of chemogenic attenuation of seizures in NHP, we believe that our results increase the translational value of this methodology, despite the limitation.

Fourth: I still disagree with the statistical approach. It remains (regardless of protests to the contrary) a violation of the assumptions of independence to analyze these data by T-Test or ANOVA. It doesn't matter if it is "generally accepted" or not - it violates the assumption of independence to treat multiple observations from a single subject as independent observations. They clearly are not. Just because others violate the statistical assumptions doesn't mean it should be allowed to continue. They either need to use a mixed model approach or an alternative approach.

R: We understand the reviewer's criticism. As suggested, we performed mixed model approach, where model selection was performed among four linear models, taking into account that subject-dependent effect:

Model 1: scaled value ~ treat * time

Model 2: scaled value ~ treat * time + (treat | subject)

Model 3: scaled value ~ treat * time + (treat | session)

Model 4: scaled value ~ treat * time + (treat | session) + (treat | subject)

where 'treat | subject' and 'treat | session' denote random effects by subject and session, respectively. The result was that Model 1 was selected with the lowest BIC, suggesting that no significant random effects related to subject or session were detected.

However, due to the small sample size, we ultimately decided to employ an alternative approach. We performed a Bayesian statistics model analysis (state-space model) to characterize the temporal dynamics of seizure amplitude and clonic seizure during pre- and post- administering DCZ or vehicle. Importantly, we implemented random effects by subject in this model, allowing us to estimate the impact of DCZ/vehicle on trends, taking subject effects into account. As shown in the new Figures 3b and 3d, the 99% credible intervals for the posterior probability of the measurement in post-DCZ and -vehicle did not overlap, suggesting that treatment effects differed significantly depending on the treatments in both cortical and clonic seizures. We believe that this new analysis meets the reviewer's request. We revised Figures 3b and 3d, the figure legend, and the Results and the Methods to reflect this change in statistical analysis.

Peer review comments,

Third round review:

Reviewer #2 (Remarks to the Author):

This is the 2nd revision of the manuscript titled "Chemogenetic attenuation of cortical seizures in primates" by Miyakawa and colleagues. The new additions to this study are an improvement to the last revision in addressing my comments and this study continues to have potential for high clinical impact and is suitable for publication in Nature Communications. There are a couple of minor points that could be addressed prior to publication.

On page 3, in the second paragraph of the introduction it states that DREADDs "can only be activated by biologically inert actuator drugs". However, this is not true as clozapine is pharmacologically active and can activate DREADDs. The supplementary figure number on page 10 last line and second line of page 11 are incorrect.

Although the authors have attempted to address the concern regarding the differences in effects between the two animals (especially since two different viral titers were used) "no clear cell-death or immune response related signal caused by DREADD expression" (page 10), this was only demonstrated pictorially in monkey 1. Additionally, in the comparison of DREADD expression between the two monkeys found in supplementary 1, the staining looks different in the shape of the immunopositive cells as well as the visualization used. Can this be addressed?

Reviewer #3 (Remarks to the Author):

The modifications to the current manuscript answered my remaining questions. I have no additional comments or questions. The revised manuscript seems to be in excellent shape.

Reviewer #4 (Remarks to the Author):

In this revision, the authors have addressed my concerns regarding the statistical approach by switching to a Bayesian framework. This is appropriate and acceptable.

However, my other comments were not adequately addressed.

First: Sedation. The authors need to discuss this limitation. There may be synergistic effects between the sedative and chemogenetics and this should be acknowledged.

Second: The lack of a statistically significant effect of bicuculline dosage is not surprising given the small number of sessions. However, the effect on clonic seizures is of borderline statistical significance. It would be useful to have these data fully reported (change in clonic seizure across different doses of bicuculline) in a supplemental table.

Third: The EEGs - I expressed concern that the effect demonstrated in the EEG recordings is not compelling. They rebutted stating,

"Although a few of the EEG traces indeed indicate that some residual ictal activity remained following DCZ treatment (e.g., Supplementary Fig. 3c) most EEG traces demonstrate that ictal activity has disappeared after DCZ (e.g., Fig. 2a; Supplementary Fig. 3ab)"

Figure 2a is not compelling. We are shown only 5 sec snippets of traces. The spectral analysis clearly shows significant elevation in power after DCZ treatment.

Figure 3a and 3b continue to show runs of fast spiking activity. Again, this isn't compelling.

"Quantitatively, we have shown that DCZ treatment significantly reduced EEG amplitudes, providing clear and compelling evidence for chemogenetic attenuation of seizure amplitude."

While I am not disputing that there is a reduction in electrographic seizure amplitude, I don't know any epileptologist that would look at these EEGs and say that seizures have been stopped. I note that they do describe this as "attenuation", which is perhaps a fair description, but from a translational perspective, attenuation is not good enough. Moreover, the fact that a second dose of DCZ did not rescue a waning effect is also concerning from a translational perspective.

REVIEWERS' COMMENTS

Reviewer #2 (Remarks to the Author):

This is the 2nd revision of the manuscript titled “Chemogenetic attenuation of cortical seizures in primates” by Miyakawa and colleagues. The new additions to this study are an improvement to the last revision in addressing my comments and this study continues to have potential for high clinical impact and is suitable for publication in Nature Communications. There are a couple of minor points that could be addressed prior to publication.

On page 3, in the second paragraph of the introduction it states that DREADDs “can only be activated by biologically inert actuator drugs”. However, this is not true as clozapine is pharmacologically active and can activate DREADDs.

R: We thank the reviewer for pointing this out. We corrected the sentence as follows:

“...are designed to be activated by biologically inert actuator drugs administered on-demand¹⁰⁻¹² (but can also be activated by clozapine, an antipsychotic drug¹³)”

The supplementary figure number on page 10 last line and second line of page 11 are incorrect.

R: We revised the supplementary figure numbers. Thank you.

Although the authors have attempted to address the concern regarding the differences in effects between the two animals (especially since two different viral titers were used) “no clear cell-death or immune response related signal caused by DREADD expression” (page 10), this was only demonstrated pictorially in monkey 1.

R: We thank the reviewer for pointing this out. We demonstrated no clear immune response related to signal caused by DREADD expression pictorially for monkey #2 as shown in Supplementary Fig 9. In this monkey, moderate neuronal loss was observed, but apparently not co-localized with DREADD expression. However, we cannot rule out the possibility that this was partly due to DREADD expression, therefore we omitted “cell-death” in the sentence.

Additionally, in the comparison of DREADD expression between the two monkeys found in supplementary 1, the staining looks different in the shape of the immunopositive cells as well as the visualization used. Can this be addressed?

R: We thank the reviewer for pointing this out. As apparent from the PET signal (Supplementary

figure 1), monkey #2 exhibited lower GFP expression which co-express with hM4Di. This seems to be reflected in the histological section. We addressed this in the Discussion. (page 13, line 12)

Reviewer #3 (Remarks to the Author):

The modifications to the current manuscript answered my remaining questions. I have no additional comments or questions. The revised manuscript seems to be in excellent shape.

R: We thank the reviewer for all the critical and constructive comments.

Reviewer #4 (Remarks to the Author):

In this revision, the authors have addressed my concerns regarding the statistical approach by switching to a Bayesian framework. This is appropriate and acceptable.

However, my other comments were not adequately addressed.

First: Sedation. The authors need to discuss this limitation. There may be synergistic effects between the sedative and chemogenetics and this should be acknowledged.

R: We thank the reviewer for pointing this out. We modified the discussion regarding the limitations of this point as follows:

“Second, the current study was performed under xylazine sedation because bodily convulsions had to be clearly distinguished from voluntary movements. Future studies are needed to directly determine the extent to which convulsions are suppressed in non-sedated animals, excluding potential synergistic effect.”

Second: The lack of a statistically significant effect of bicuculline dosage is not surprising given the small number of sessions. However, the effect on clonic seizures is of borderline statistical significance. It would be useful to have these data fully reported (change in clonic seizure across different doses of bicuculline) in a supplemental table.

R: According to the advice, we reported normalized clonic seizure magnitude, which we used for the statistics, in Supplementary Table 1.

Third: The EEGs - I expressed concern that the effect demonstrated in the EEG recordings is not compelling. They rebutted stating,

"Although a few of the EEG traces indeed indicate that some residual ictal activity remained following DCZ treatment (e.g., Supplementary Fig. 3c) most EEG traces demonstrate that ictal activity has disappeared after DCZ (e.g., Fig. 2a; Supplementary Fig. 3ab)"

Figure 2a is not compelling. We are shown only 5 sec snippets of traces. The spectral analysis clearly shows significant elevation in power after DCZ treatment.

Figure 3a and 3b continue to show runs of fast spiking activity. Again, this isn't compelling.

"Quantitatively, we have shown that DCZ treatment significantly reduced EEG amplitudes, providing clear and compelling evidence for chemogenetic attenuation of seizure amplitude."

While I am not disputing that there is a reduction in electrographic seizure amplitude, I don't know any epileptologist that would look at these EEGs and say that seizures have been stopped. I note that they do describe this as "attenuation", which is perhaps a fair description, but from a translational perspective, attenuation is not good enough. Moreover, the fact that a second dose of DCZ did not rescue a waning effect is also concerning from a translational perspective.

R: We appreciate the reviewer's further clarification of the comments. We now understand the point. We acknowledge that the DCZ application did not fully block seizures in our current study. But to be fair, this was not peculiar to the present case with monkeys, but was also the case in many of previous studies with rodents. We agree that this is an important issue that needs improvement for future clinical application and have mentioned it in the Discussion (page 15 line 3).